# On the timescale of drought indices for monitoring streamflow drought considering catchment hydrological regimes

Oscar M. Baez-Villanueva[1,2], Mauricio Zambrano-Bigiarini[3,4], Diego G. Miralles[1], Hylke E. Beck[5], Jonatan F. Siegmund[6], Camila Alvarez-Garreton[4], Koen Verbist[7], René Garreaud[8,4], Juan Pablo Boisier[8,4], and Mauricio Galleguillos[9,4]

[1]Hydro-Climate Extremes Lab (H-CEL), Ghent University, Ghent, Belgium
[2]Institute for Technology and Resources Management in the Tropics and Subtropics (ITT), TH Köln, Cologne, Germany
[3]Department of Civil Engineering, Universidad de la Frontera, Temuco, Chile
[4]Center for Climate and Resilience Research, Universidad de Chile, Santiago, Chile
[5]Climate and Livability Initiative, Physical Sciences and Engineering, King Abdullah University of Science and Technology, Thuwal, Saudi Arabia
[6]Ernst Young GmbH, Wirtschaftsprüfungsgesellschaft, Stuttgart, Germany
[7]UNESCO International Hydrological Programme, Paris, France
[8]Department of Geophysics, Universidad de Chile, Santiago, Chile
[9]Faculty of Engineering and Sciences, Universidad Adolfo Ibáñez, Santiago, Chile

**Correspondence:** Mauricio Zambrano-Bigiarini (mauricio.zambrano@ufrontera.cl)

**Abstract.** There is a wide variety of drought indices, yet a consensus on suitable indices and temporal scales for monitoring streamflow drought remains elusive across diverse hydrological settings. Considering the growing interest in spatially-distributed indices for ungauged areas, this study addresses the following questions: $i)$ what temporal scales of precipitation-based indices are most suitable to assess streamflow drought in catchments with different hydrological regimes?, $ii)$ do soil moisture indices outperform meteorological indices as proxies for streamflow drought?, $iii)$ are snow indices more effective than meteorological indices for assessing streamflow drought in snow-influenced catchments? To answer these questions, we examined one hundred near-natural catchments in Chile with four hydrological regimes, using the Standardised Precipitation Index (SPI), Standardised Precipitation and Evapotranspiration Index (SPEI), Empirical Standardised Soil Moisture Index (ES-SMI), and standardised Snow Water Equivalent Index (SWEI), aggregated across various temporal scales. Cross-correlation and event coincidence analysis were applied between these indices and the Standardised Streamflow Index at a temporal scale of one month (SSI-1), as representative of streamflow drought events. Our results underscore that there is not a single drought index and temporal scale best suited to characterise all streamflow droughts in Chile, and their suitability largely depend on catchment memory Specifically, in snowmelt-driven catchments characterised by a slow streamflow response to precipitation, the SPI at accumulation periods of 12–24 months serves as the best proxy for characterising streamflow droughts, with median correlation and coincidence rates of approximately 0.70–0.75 and 0.58–0.75, respectively. In contrast, the SPI at a 3-month accumulation period is the best proxy over faster-response rainfall-driven catchments, with median coincidence rates of around 0.55. Despite soil moisture and snowpack being key variables that modulate the propagation of meteorological deficits into hydrological ones, meteorological indices are better proxies for streamflow drought. Finally, to exclude the influence of

non-drought periods, we recommend using the event coincidence analysis, a method that helps assessing the suitability of meteorological, soil moisture, and/or snow drought indices as proxies of streamflow drought events.

## 1  Introduction

A wide variety of drought indices coexists today (e.g., Svoboda et al., 2002; Hayes et al., 2012; Cammalleri et al., 2021) without consensus on which are more appropriate for monitoring streamflow drought, and over which temporal scales (Vicente-Serrano and López-Moreno, 2005; Jain et al., 2015; Bachmair et al., 2016; Barker et al., 2016; Zhao et al., 2017; Slette et al., 2019; Hoffmann et al., 2020; Vorobevskii et al., 2022). Primarily because just a few of them have truly been tested against drought impact data (Blauhut et al., 2015). Drought indices are often used alone or in combination to operationally monitor the onset, duration, and severity of drought events. Examples include the U.S. Drought Monitor (Svoboda et al., 2002; Leeper et al., 2022), the European Drought Observatory (Vogt, 2011; Cammalleri et al., 2021), the German Drought Monitor (Zink et al., 2016), the South Asia Drought Monitor (Saha et al., 2021), the Australian Drought Monitor (Cobon et al., n.d.), and the Global Drought Monitor Portal (Heim Jr. and Brewer, 2012; Hao et al., 2017). The low agreement among these indices, as well as the limited knowledge about their critical thresholds related to tangible impacts on society, economy, and ecosystems (Steinemann, 2003; Lac; Bachmair et al., 2015; Stagge et al., 2015a; Torelló-Sentelles and Franzke, 2022), hinders their effective use for decision-making purposes, drought mitigation, and adaptation (Steinemann and Cavalcanti, 2006; Steinemann, 2014; Steinemann et al., 2015). Before selecting a drought index, proper case-specific understanding is essential. This understanding enables effective monitoring, management, and planning, and can lead to developing mitigation strategies that ensure the resilience of our environment and communities (Bachmair et al., 2016).

Drought events are frequently classified as meteorological, soil moisture, and hydrological (Haile et al., 2020), each with different characteristics, impacts, and also with different indices used to monitor their development. Well-know meteorological drought indices include percent of normal (Zargar et al., 2011), deciles (Gibbs and Maher, 1967), Z-score (ZSI; Palmer, 1965a; Zargar et al., 2011), Rainfall Anomaly Index (RAI; van Rooy, 1965; Keyantash and Dracup, 2002), Palmer Drought Severity Index (PDSI; Palmer, 1965b), Standardised Precipitation Index (SPI; McKee et al., 1993), Effective Drought Index (EDI; Byun and Wilhite, 1999), and Standardised Precipitation Evapotranspiration Index (SPEI; Vicente-Serrano et al., 2010).

Although meteorological and soil moisture drought indices are relatively easy to calculate due to either their reliance on frequently-measured precipitation ($P$) or their reduced number of input variables, most of the socio-ecological impacts of droughts are related to surface water and groundwater deficits (altogether referred to as hydrological droughts), i.e., water shortages in rivers, lakes, and groundwater (Van Loon, 2015). Likewise, numerous indices have been used to monitor soil moisture drought, including the Palmer Moisture Anomaly Index (Z-index; Karl, 1986; Keyantash and Dracup, 2002), the Soil Moisture Deficit Index (SMDI; Narasimhan and Srinivasan, 2005), and the Empirical Standardised Soil Moisture Index

(ESSMI; Carrão et al., 2016). Drought indices describing hydrological droughts use a variety of variables related to surface and groundwater amounts. Examples are: the Standardised Streamflow Index (SSI; Vicente Serrano et al., 2012) and the Standardised Runoff Index (SRI; Shukla and Wood, 2008)) both based on streamflow ($Q$); the Surface Water Supply Index (SWSI; Shafer and Dezman, 1982) based on snow water equivalent ($SWE$) as representative of the snowpack, $Q$, reservoir storage, and $P$; the Standardised Groundwater level Index (SGI; Bloomfield and Marchant, 2013a; Van Loon et al., 2017)

based on groundwater levels or terrestrial water storage; the Standardised Snow Melt and Rain Index (SMRI; Staudinger et al., 2014) based on snow storage and $P$; and the Standardised Snow Water Equivalent Index (SWEI; Huning and AghaKouchak, 2020) based on $SWE$. As these variables are frequently less easily accessible, streamflow drought indices are often used to characterise hydrological droughts (Barker et al., 2016; Wu et al., 2017; Peña-Gallardo et al., 2019; Bhardwaj et al., 2020).

The propagation of meteorological drought into streamflow drought is a complex process modulated by the land component

of the hydrological cycle. This propagation may lead to clustered, attenuated, delayed and prolonged events, which are mediated by catchments characteristics, climate conditions, and soil moisture dynamics (Van Loon, 2015). The combination of these factors determine the hydrological memory of a basin, which modulates the cumulative effects of precipitation deficits on the severity, duration and recovery of droughts (Alvarez-Garreton et al., 2021). Various approaches have been used to evaluate how $P$ deficits propagate through the hydrological cycle into $Q$ deficits (Rahiz and New, 2013; Haslinger et al., 2014; Zhu

et al., 2016; Apurv et al., 2017a; Gevaert et al., 2018; Jehanzaib et al., 2020), and studies suggest that combining drought indices based on different variables could provide a stronger agreement with ground-based hydrological impacts (Niemeyer et al., 2008; Sivakumar et al., 2011; Zargar et al., 2011). However, there is still a need to identify which indices and time scales are most appropriate for assessing streamflow droughts, and how this choice varies as a function of the specific hydrological regime. This is particularly important in catchments with a sparse monitoring network, where the identification and assessment

of streamflow droughts can be challenging.

Therefore, this study aims to answer the following questions: $i$) what temporal scales of (the frequently used and easy to calculate) SPI and SPEI meteorological indices can be used as proxies for streamflow drought in catchments with different hydrological regimes? $ii$) considering that the soil acts as a natural reservoir to maintain streamflow during periods of reduced $P$, can a soil moisture drought index be used to assess streamflow droughts instead of the SPI and SPEI meteorological indices?;

and $iii$) considering that snowmelt is an important moisture contribution to streamflow and surface water availability during spring and summer months in catchments with a pronounced snow influence, can a $SWE$ index be used to assess streamflow droughts instead of the SPI and SPEI in those catchments? Answering these questions can help us to design effective monitoring and early warning systems, to elaborate better drought mitigation strategies (Huang et al., 2017) and management practices to reduce societal vulnerability to drought (Svoboda et al., 2002), and to better understand how future changes in meteorological

variables will impact on streamflow droughts.

## 2 Study area and selected catchments

Continental Chile spans 4,300 km of latitudinal extension (17.5°–56.0°S) but only 200 km of longitudinal extension on average (76.0°–66.0°W). It is bounded to the north by Peru, to the west by the Pacific Ocean, and to the east by Bolivia and Argentina, with elevations ranging from 0 to 6,892 m a.s.l. in the Andes cordillera. Figure 1 shows the elevation (Jarvis et al., 2008), the land cover classification (Zhao et al., 2016), the Köppen-Geiger climate classification (Beck et al., 2018), and the 100 near-natural catchments with different hydrological regimes that are used in this study (same as in Baez-Villanueva et al., 2021). The climate transitions from arid in the north to temperate and humid climates in the Central Chile and South regions. In general, $P$ increases with latitude (in the southern direction) and elevation. Likewise, $Q$ also tends to increase from north to south (Alvarez-Garreton et al., 2018; Vásquez et al., 2021). The country is home of about 18 million inhabitants, two thirds of them concentrated in central Chile (30°–38°S), region where intensive agriculture and other economic activities prevail.

The El Niño-Southern Oscillation (ENSO) influences winter $P$ in Central Chile, producing positive and negative anomalies during El Niño and La Niña events, respectively (Montecinos and Aceituno, 2003; Verbist et al., 2010; Robertson et al., 2014). Recently, Central Chile experienced a prolonged period of dry conditions from 2010 onward, known as the Central Chile *megadrought* (Garreaud et al., 2017b), with $P$ deficits of 25–80% combined with an unprecedented increase in evaporative demand, which affected approximately 70% of the Chilean population. These conditions had adverse effects on snowpack, groundwater, and surface water levels, with a decrease in mean river discharge up to 90% (Garreaud et al., 2020) and up to 60% increase in areas affected by forest fires (González et al., 2018).

Boisier et al. (2016) found that large-scale circulation changes and the Pacific Decadal Oscillation explains around 50% of the $P$ decline observed in Central Chile, while the Antarctic stratospheric ozone depletion has played a major role in the summer $P$ decline Boisier et al. (2018b). On the other hand, Garreaud et al. (2017a) analysed the extraordinary character of the megadrought using century long historical records and a millennial tree-ring reconstruction of regional $P$, along with describing its impacts on regional hydroclimate and vegetation, while Garreaud et al. (2019) found that the exceptional length of the megadrought is due to the prevalence of a circulation dipole-hindering the passage of extratropical storms over Central Chile.

We carefully selected a subset of 100 near-natural catchments, as detailed in Baez-Villanueva et al. (2021), aiming to isolate the influence of natural hydrological processes from human interventions. By selecting catchments with minimal human alterations, our goal was to more clearly understand the intrinsic relationships between meteorological, soil moisture, and snow drought indices and streamflow dynamics in a predominantly natural context. The selected catchments met the following criteria:

1. *Less than 25% of missing values in the daily Q time series (which could be non-consecutive).*

2. *Absence of large dams.*

3. *Less than 10% of Q allocated to consumptive uses (intervention degree < 10%).*

4. *Catchments not dominated by glaciers (glacier area < 5%).*

5. *Urban area less than 5%.*

6. *Minimal irrigation abstractions (agriculture fraction < 20%).*

7. *Less than 20% of the area covered by forest plantations.*

8. *No signs of artificial regulation in the hydrograph (resulting in the exclusion of 10 catchments).*

These selected catchments vary in size (35 to 11,137 km$^2$, median area of 645 km$^2$), dominant Köppen-Geiger climate type (from BWh, BWk, BSh, BSk or ET), annual precipitation (from 56.58 to 3,914.26 mm y$^{-1}$), aridity index (from 0.28 to 9.32), dominant land cover type (shrublands, grasslands, barren land, or native forests), and main geological characteristics. They are part of the Catchment Attributes and Meteorology for Large-sample Studies dataset in Chile (CAMELS-CL; Alvarez-Garreton et al. (2018)), and their attributes are summarised in the Supplementary Material. As explained in (Baez-Villanueva et al., 2021), we classified these catchments into four hydrological regimes: $i$) nival (snow-dominated), $ii$) nivo-pluvial (snow-dominated with rain influence), $iii$) pluvio-nival (rain-dominated with snow influence), and $iv$) pluvial (rain-dominated regime). The CAMELS-CL dataset provides extensive information on 516 Chilean catchments, including hydroclimatic indices, geomorphological characteristics, location, topography, geology, soil types, land cover, hydrological signatures, and the degree of human intervention. These hydrological regimes were derived by analysing $P$ and $Q$ data, and the timing of $Q$ peaks relative to maximum $P$. Finally, the derived hydrological regimes were compared against other Studies (Baez-Villanueva et al., 2021). The nival hydrological regime characterises catchments dominated by snowfall, where the peak $Q$ occurs in spring or summer due to snowmelt. In contrast, the pluvial regime characterises catchments dominated by rainfall, where $Q$ is driven by rainfall-runoff processes.

Classifying basins by their hydrological regime is important to understand and characterise drought propagation. Recently, Alvarez-Garreton et al. (2021) found that annual $P$ anomalies during the megadrought have been larger in nival catchments compared to pluvial catchments, with nival catchments being more prone to an intensified propagation of persistent droughts. This was attributed to the accumulation of precipitation deficits on basins with longer streamflow response time to precipitation (refer as catchment memory), compared to fast response pluvial basins. For a detailed representation of conceptual hydrographs associated with each hydrological regime, please refer to Figure S1 of the Supplementary Material.

## 3  Data and methods

To address the research questions outlined in Section 1, we followed a three-step procedure, as illustrated in Figure 2: $i$) computing meteorological (SPI, SPEI), soil moisture (ESSMI), and snow-related (SWEI) drought indices; $ii$) applying cross-correlation and event coincidence analyses among the previous drought indices and the SSI at temporal scale of 1 month as representative of streamflow deficit conditions, within the study catchments; and finally $iii$) evaluating the cross-correlation and event coincidence analysis results considering the hydrological regime of the selected catchments.

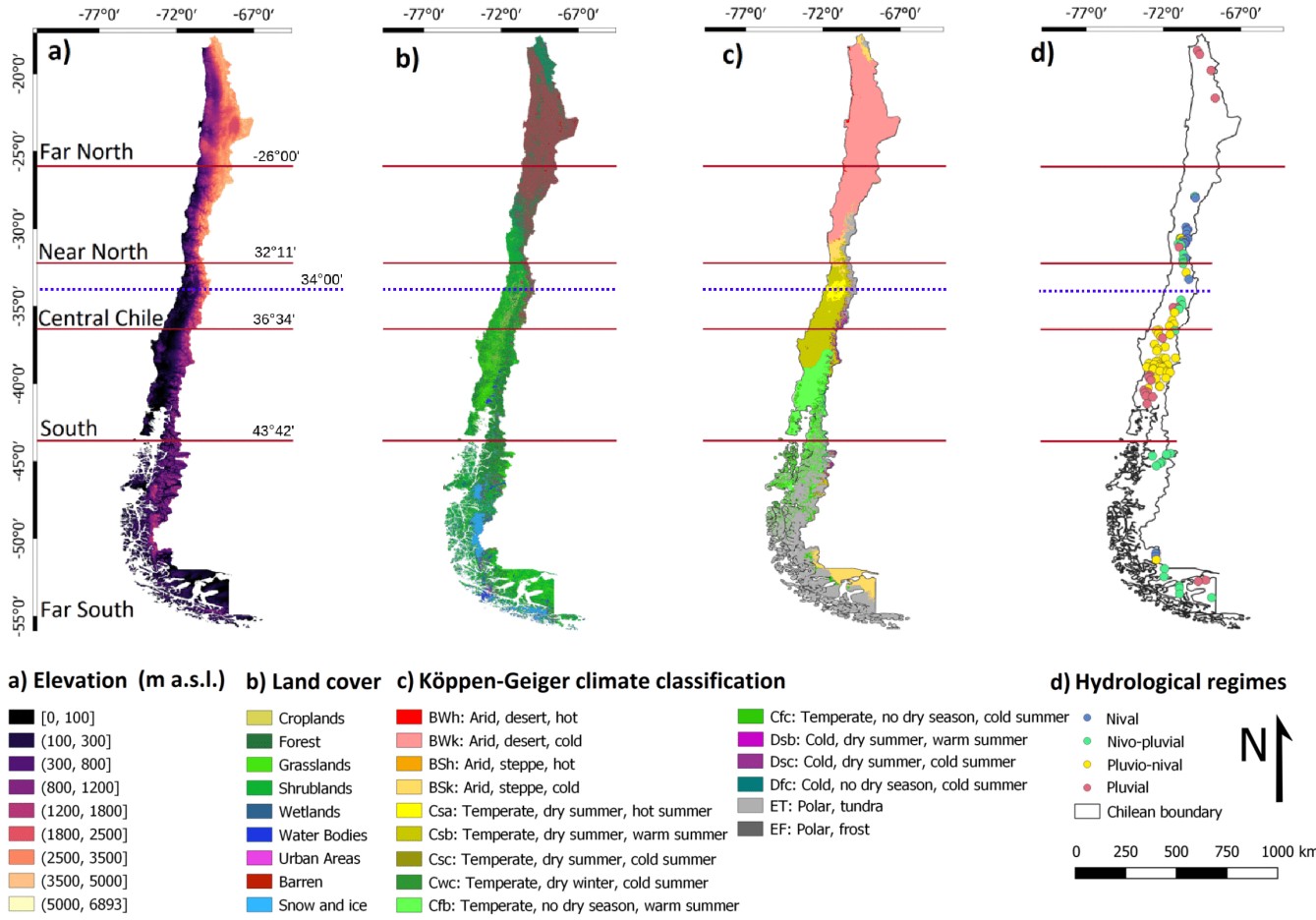

**Figure 1.** Study area adapted from Baez-Villanueva et al. (2021): $a$) elevation (SRTMv4.1; Jarvis et al., 2008) and major macroclimatic zones of Chile: $i$) Far North (17.50–26.00°S), $ii$) Near North (26.00–32.18°S), $iii$) Central Chile (32.18–36.40°S), $iv$) South (36.40–43.70°S), and $v$) Austral/Far South (43.70–56.00°S); $b$) land cover classification (Zhao et al., 2016); $c$) Köppen-Geiger climate classification (Beck et al., 2018); and $d$) 100 near-natural catchments with different hydrological regimes (Baez-Villanueva et al., 2021).

## 3.1 Gridded datasets

Table 1 summarises the gridded datasets used in this study and their related variables. CR2METv2.5 (Boisier et al., 2018a) provides daily gridded estimates of $P$ and maximum and minimum temperature ($T$) over continental Chile at a spatial resolution of 0.05° from 1979 to near present. $P$ estimates are computed by combining ground-based observations with reanalysis data from ERA5, while $T$ is estimated using multivariate regression from Moderate Resolution Imaging Spectroradiometer (MODIS) land surface temperature (LST) and ERA5 as covariates (Alvarez-Garreton et al., 2018; Boisier et al., 2018a). The Hargreaves-Samani equation (Hargreaves and Samani, 1985) was used to obtain daily potential evaporation ($PE$) from CR2METv2.5

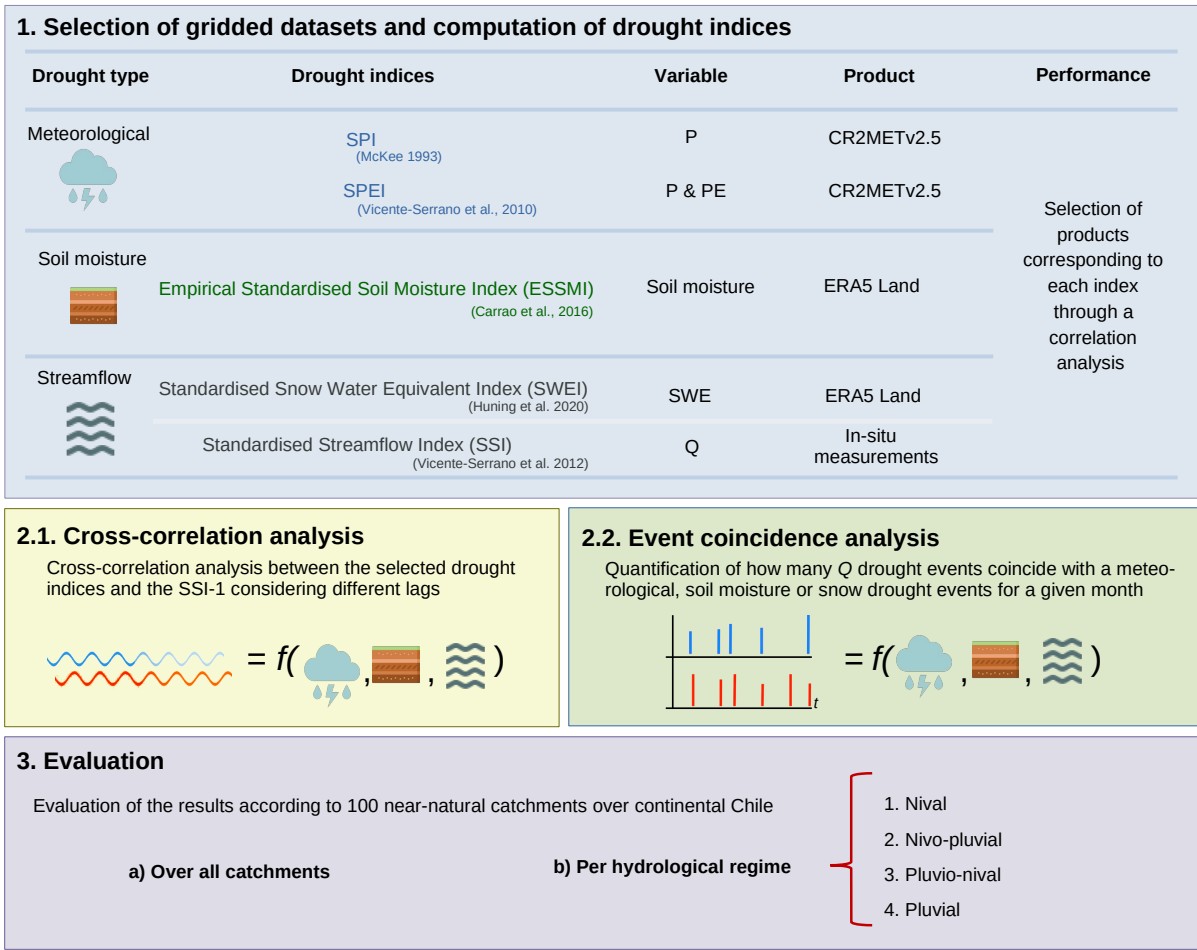

**Figure 2.** Schematic summarising the three-step approach to answer the research questions presented in Section 1.

maximum and minimum daily $T$ at the same $0.05°$ spatial resolution. We selected CR2METv2.5 due to its strong performance compared to other products in previous studies (Baez-Villanueva et al., 2020, 2021).

ERA5-Land (Muñoz-Sabater et al., 2021) is an enhanced hourly global dataset for the land component of ERA5 that de-
155 scribes the evolution of energy and water cycles consistently over land. This product was developed through numerical inte-
grations of the ECMWF land surface model driven by the downscaled meteorological forcing from ERA5, which included an
elevation correction for the thermodynamic near-surface state (Muñoz-Sabater et al., 2021). ERA5-Land has a spatial resolu-
tion of $0.10°$ and spans from 1950 to the present. The volumetric soil water in this product is associated with soil texture, soil
depth, and the underlying groundwater level. Additionally, ERA5-Land also provides snow depth as $SWE$.
Other soil moisture and $SWE$ datasets were also evaluated and deemed less adequate for the study (see Section 1 of the
Supplementary Material).

**Table 1.** Gridded hydrometeorological products used in this study.

| Variable | Product | Period | Spatio-temporal resolution | References |
|----------|---------|--------|----------------------------|------------|
| $P$ | CR2METv2.5 | 1979–present | 0.05°; daily | Boisier et al. (2018a) |
| $T_{max}$ and $T_{min}$ | CR2METv2.5 | 1979–present | 0.05°; daily | Boisier et al. (2018a) |
| Soil moisture | ERA5-Land | 1940–present | 0.10°; hourly | Muñoz-Sabater et al. (2021) |
| $SWE$ | ERA5-Land | 1950–present | 0.10°; hourly | Muñoz-Sabater et al. (2021) |

## 3.2 Drought Indices

### 3.2.1 Standardised Precipitation Index

The Standardised Precipitation Index (SPI; McKee et al., 1993) is considered the standard for monitoring and characterising meteorological drought at different time scales (Hayes et al., 2011; WMO, 2012) and has been widely used for decades (e.g., Wu et al., 2001; Zhang et al., 2013; Meroni et al., 2017; Liu et al., 2021) including Chile (Zambrano et al., 2017; Zambrano-Bigiarini and Baez-Villanueva, 2019). The SPI compares the accumulated $P$ for a specific temporal scale at a given location with its long-term (e.g., 30 years) distribution, making it useful for comparing $P$ deficits or excesses across regions with different climates. The SPI follows a three-step approach: $i$) selecting a probability density function (PDF) and a method for estimating its parameters (e.g., Hosking and Wallis, 1995) to represent the long-term distribution of accumulated $P$ over the desired period; $ii$) obtaining the cumulative distribution function (CDF) from the fitted distribution; and $iii$) transforming accumulated $P$ into a normal distribution with a mean of zero and standard deviation of one, using an equi-percentile inverse transformation to derive the SPI values. The SPI represent the number of standard deviations that cumulative $P$ deviates from its long-term average (JRC, 2011).

In this study, the SPI was calculated at different scales (1, 3, 6, 9, 12, 18, and 24 months) using CR2METv2.5. The selected $P$ accumulation periods were adjusted through the two-parameter gamma distribution, as recommended by Stagge et al. (2015b), in combination with the unbiased Probability Weighted Moments (Hosking and Wallis, 1995) to estimate the parameters of the gamma distribution.

### 3.2.2 Standardised Precipitation and Evapotranspiration Index

The Standardised Precipitation and Evapotranspiration Index (SPEI; Vicente-Serrano et al., 2010; Beguería et al., 2014) is a multi-scalar index similar to the SPI. However, instead of using only $P$ as input, the SPEI uses the climatic water balance given by $P$ minus potential evaporation ($P - PE$). The SPEI is suited to identifying, tracking, and exploring how drought conditions are affected by global warming, e.g., it allows to robustly consider the influence of recent positive trends in the evaporative demand on the temporal evolution of streamflow (Vicente-Serrano et al., 2010, 2014; Liu et al., 2021). To calculate this index, the often highly skewed distribution of the climatic water balance is transformed to the standard normal distribution using the univariate log-logistic probability distribution (Vicente-Serrano et al., 2010, 2014). Temporal scales used to calculate

this index are similar to those of SPI, and share the same relation with monitoring water availability. In this study, the SPEI was calculated at different scales (1, 3, 6, 9, 12, 18, and 24 months). The selected accumulation periods were adjusted through the log-logistic distribution as recommended by Vicente-Serrano et al. (2010), in combination with the unbiased Probability Weighted Moments method (Hosking and Wallis, 1995) to estimate the parameters of the log-logistic distribution.

### 3.2.3   Empirical Standardised Soil Moisture Index

Soil moisture modulates both the land surface water and energy cycles, and influences plant productivity and yields, playing a key role in agriculture and water supply. Among the many indices available for monitoring soil moisture deficits (e.g. Sims et al., 2002; Zargar et al., 2011; Wang et al., 2015), we chose the Empirical Standardised Soil Moisture Index (ESSMI; Carrão et al., 2016), a spatially-invariant probabilistic index designed to detect soil moisture anomalies. The ESSMI was chosen because its non-parametric nature has proven to be more accurate than previously proposed parametric indices (Carrão et al., 2016). The computation of the ESSMI was carried out by fitting a non-parametric empirical probability density function (ePDF) to historical time-series of SM, using a kernel density estimation (Silverman, 1986) optimised for each grid-cell, and then transforming it into a normal distribution with a mean of zero and standard deviation of one. As for the SPI and SPEI, negative standard normal values indicate dry conditions, whereas positive values indicate wet conditions. Typical accumulation periods used for ESSMI are 1, 3, 6, or 12 months. We used soil moisture data of ERA5-Land after evaluating other products such as ERA5, SMAP-L3E and SMAP-L4SM (see Section 1 of the Supplementary Material).

### 3.2.4   Standardised Snow Water Equivalent Index

The Standardised Snow Water Equivalent Index (SWEI; Huning and AghaKouchak, 2020) aims at quantifying snow drought conditions. To compute this index we followed Huning and AghaKouchak (2020) who used a non-parametric approach (Farahmand and AghaKouchak, 2015) to standardise $SWE$ time series. In general, instead of fitting a specific distribution function to the data, the SWEI computes the probabilities associated with the $SWE$ time series using the empirical Gringorten plotting position (Gringorten, 1963):

$$p(A_{m,i}) = (i - 0.44)/(N + 0.12) \tag{1}$$

where $i$ is the rank of the non-zero variable (in increasing order), and $N$ is the sample size. These ranks are determined using the 3-month $SWE$ integration for any given month ($A_{m,i} = SWE_m + SWE_{m-1} + SWE_{m-2}$). In this sense, $A_{m,i}$ provides an integrated measure of the $SWE$ amount and its persistence over each value. Afterwards, $A_{m,i}$ is standardised by transforming the empirical probability $p$ to the standard normal distribution:

$$SWEI(A_{m,i}) = \phi^{-1}[p(A_{m,i})] \tag{2}$$

While this index has the potential to be applied across various time scales, its interpretation differs from that of other drought-related indices. This distinction is related to the presence of seasonal snowpack in specific regions, where several months during the warm season may exhibit a lack of $SWE$. The absence of $SWE$ hinders the use of the SWEI during those months. As

a result, Huning and AghaKouchak (2020) recommended to $i$) exclude the warm season months (in this case, November to April) to prevent potential biases in the analysis of the SWEI, and $ii$) use accumulation periods shorter than 6 months for SWEI.

Therefore, we computed the SWEI for accumulation periods of 1, 3, and 6 months, assuming that the selected $SWE$ dataset offers valuable insights into the spatial and temporal distribution of snow. Additionally, we filtered out grid cells with very low $SWE$ ($<1$ mm) to avoid introducing bias in the calculation of the index.

### 3.2.5 Standardised Streamflow Index

Streamflow drought indices directly explain the consequences of climatic anomalies on current hydrological conditions, in contrast to climate-based indices that describe climate anomalies in isolation from their hydrological context (Shukla and Wood, 2008). The SSI follows the same idea of the SPI: $i$) selecting a PDF and estimating its parameters to model the long-term distribution of $Q$ for a given period, and $ii$) deriving the corresponding CDF to transform $Q$ into a standard normal distribution. Here, we used the SSI with a one-month accumulation period (SSI-1) to characterise streamflow drought because: $i$) it integrates catchment-scale hydrogeological processes (Barker et al., 2016) and the hydrometeorology (Stahl et al., 2020) from the preceding months; $ii$) while the SSI-1 may be susceptible to short-term fluctuations (to a lesser extent than SPI-1 and SPEI-1), its selection facilitates a direct comparison between the current month's streamflows and the accumulated values of selected hydrological cycle variables ($P$, $P-PE$, $SM$, and $SWE$) at various temporal scales; and $iii$) it has been widely used in previous studies on streamflow droughts (Barker et al., 2016; Li et al., 2020; Stahl et al., 2020; Wang et al., 2020; Bevacqua et al., 2021).

Due to the important variability in the statistical properties of monthly series of climatic water balance, several probability distributions can be used to compute a reliable SSI (Vicente Serrano et al., 2012). Here, we use the GEV distribution to calculate the SSI, as it was one of the recommended distributions by Vicente Serrano et al. (2012), besides the log-logistic distribution. Quality-controlled $Q$ data were obtained for each catchment from the Center for Climate and Resilience Research (CR2; http://www.cr2.cl/datos-de-caudales/).

### 3.3 Evaluation of proxies of streamflow drought

For evaluating the ability of the selected drought indices to be used as proxies for streamflow drought, we employed two analyses in each one of the 100 near-natural catchments: $i$) a cross-correlation analysis, to measure the similarity between the time series of the selected drought indices (SPI, SPEI, ESSMI, SWEI) at different temporal scales and the SSI-1, and $ii$) an event coincidence analysis, to evaluate the statistical interdependence between specific events from the selected indices and SSI-1. Both analysis were performed for the period 1979–2020, using lag times from 0 to 12 months between the analysed drought index and SSI-1.

Results of the cross-correlation and event coincidence analysis for the selected 100 near-natural catchments were evaluated both for all the catchments at once, and for the catchments grouped by hydrological regime. The analysed hydrological regimes are: $i$) nival (16 catchments), $ii$) nivo-pluvial (nival catchments with a rain component: 25 catchments), $iii$) pluvio-nival,

(pluvial catchments with a snow component: 40 catchments), and $iv$) pluvial (19 catchments), as shown in Figure 1d. These hydrological regimes were determined according to the contribution of solid and liquid $P$ to the mean monthly $Q$ values as described in (Baez-Villanueva et al., 2021). Figure S1 of the Supplementary Material, adapted from Baez-Villanueva et al. (2021), displays conceptual hydrographs for each one of these hydrological regimes.

### 3.3.1 Cross-correlation analysis

A cross-correlation analysis (Equation 3) measures the similarity of two time series as a function of the relative displacement of one relative to the other. Two time series ($X_t$ and $Y_t$) may be related at different lags between each other (e.g., the time series $Y_t$ may be related to past or future lags of $X_t$). The cross-correlation function is defined as a set of correlations between $X_{t+h}$ and $Y_t$ for different lags (h = 0, $\pm$1, $\pm$2, ... ,$\pm$n). In this study, $X_t$ represents the time series of the SSI-1 for each one of the 100 selected catchments, while $Y_t$ represents the time series of SPI, SPEI, ESSMI, and SWEI, alternatively. This analysis is useful for identifying whether $Y_t$ could be useful to predict $X_t$. Therefore, we used positive lag values from zero (cross-correlation analysis at the same period for both variables) to 12 months assuming that the variable $X$ lags the variable $Y$. In this sense, $X$ lags $Y$ means that changes in the selected drought indices (the SPI, SPEI, ESSMI, and SWEI) precede changes in the SSI-1; and therefore, variations in $P$, $PE$, soil moisture, and $SWE$ might have a delayed impact on $Q$. The cross-correlation analysis have been previously applied to analyse the propagation from meteorological to streamflow droughts and investigate the influence of climate and catchment properties on streamflow drought characteristics (Barker et al., 2016; Peña-Gallardo et al., 2019).

$$r_h = \frac{\sum_{t=1}^{n} \left[ (Y_t - \bar{Y})(X_{t+h} - \bar{X}) \right]}{\sqrt{\sum_{t=1}^{n}(Y_t - \bar{Y})^2}\sqrt{\sum_{t=1}^{n}(X_{t+h} - \bar{X})^2}} \tag{3}$$

where $\bar{X}$ represents the mean value of the SSI-1, and $\bar{Y}$ represents the mean value of the SPI, SPEI, ESSMI, and SWEI time series, alternatively.

### 3.3.2 Event coincidence analysis

The cross-correlation analysis uses the whole time series of the selected drought indices and, therefore, takes into account the influence of non-drought periods. To overcome the previous limitation of the cross-correlation analysis, we also use the event coincidence analysis in this work (Donges et al., 2016; Siegmund et al., 2017), a statistical technique specifically designed to assess the inter-dependency between two event time series; in this case, those of the meteorological, soil moisture or snow drought index, and the SSI-1. This method examines the degree of temporal association between events by assessing the frequency of simultaneous events, considering delayed responses (specified by a lag term $\tau$) and uncertain timing (specified by a coincidence window $\Delta T$) between binary time series (Siegmund et al., 2017). The binary series are computed using a threshold, where entries with a value of one correspond to time steps with an event, and entries with zero correspond to time steps without an event (Donges et al., 2016). This method is able to quantify the strength of statistical interrelationships

between two event series A and B, addressing B-type events as precursors (the event B precedes event A) and triggers (the event B triggers the event A). We analysed only precursor coincidence rates in this study, and not trigger rates, because we were interested in determining the number of streamflow drought events preceded by or coinciding with abnormal meteorological, soil moisture, and snow drought events, and not the number of streamflow droughts that were triggered by meteorological, soil moisture, or snow droughts. The precursor coincidence rate is calculated as follows:

$$r_p(\Delta T, \tau) = \frac{1}{N_A} \sum_{i=1}^{N_A} \Theta \left[ \sum_{j=1}^{N_B} 1_{[0,\Delta T]}((t_i^A - \tau) - t_j^B) \right] \tag{4}$$

Equation 4 quantifies the proportion of A-type events that are preceded by at least one B-type event. It considers multiple B-type events within the coincidence interval as a single occurrence. The Heaviside function $\Theta(\cdot)$ is used in the equation, defined as $\Theta(x) = 0$ for $x \leq 0$ and $\Theta(x) = 1$, otherwise. Additionally, the indicator function $1_I(\cdot)$ is employed to represent the presence of values within a specific interval $I$, defined as $1_I(x) = 1$ for $x$ within $I$ and $1_I(x) = 0$ otherwise. The optimal value of the precursor coincidence rate is one and represents the ratio of events A that coincide with events B. For example, if a value of 0.6 is obtained in this case, it means that 60% of streamflow droughts align with a drought event identified using the chosen index.

This method, has been widely used to analyse $i$) floods as triggers of epidemic outbreaks (Donges et al., 2016); $ii$) meteorological extremes (Siegmund et al., 2016); $iii$) the timing of droughts and floods (Siegmund et al., 2017); and $iv$) droughts and wet spells (He and Sheffield, 2020), and to calculate precursor and trigger coincidence rates (Donges et al., 2016). Here, we use independently two different thresholds to define an event (Lloyd-Hughes and Saunders, 2002; Barker et al., 2016): $i$) a value $\leq -1.0$ to evaluate moderate droughts, and $ii$) a value $\leq -1.5$ to evaluate severe droughts. We excluded extreme droughts (values $\leq -2.0$) as not all indices presented standardised values below $-2.0$. In this sense, values lower than or equal to the selected threshold are considered as events (value $= 1$), while values greater than the threshold represent periods without an event (value $= 0$).

Additionally, to assess the strength of the statistical relationship between these drought types, we performed a significance test assuming a Poisson process, which quantifies the robustness of the relationship between them. This test evaluates the hypothesis that the occurrence of these signals is distributed randomly and not related to each other. The methodology for this approach is explained in detail in Donges et al. (2016) and has been used in previous studies (Donges et al., 2016; Siegmund et al., 2017; He and Sheffield, 2020). A schematic representation of the event coincidence analysis is presented in Figure 3.

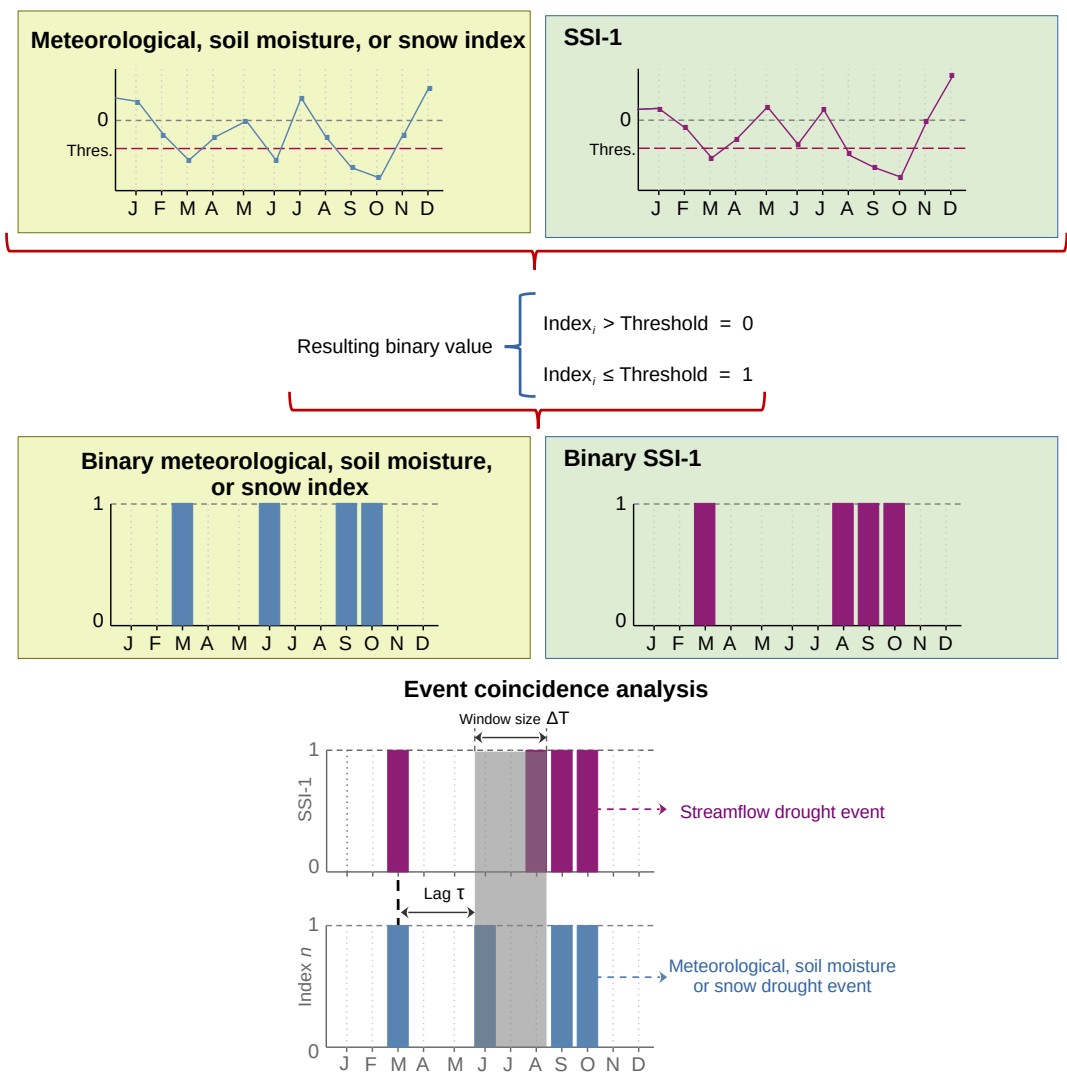

**Figure 3.** Schematic of the event coincidence analysis between meteorological, soil moisture, or snow drought indices and the SSI-1 considering a time lag $\tau$ with a certain window $\Delta$T.

## 4 Results and discussion

### 4.1 Proxies of streamflow drought: all catchments

This section analyses the cross-correlation values and precursor coincidence rates between the selected drought indices (SPI, SPEI, ESSMI, SWEI) and the SSI-1 when all the 100 near-natural catchments are analysed at once. An arbitrary lower threshold of 0.4 was used for cross-correlation values and precursor coincidence rates to separate drought indices and temporal scales that could be considered a good proxy of streamflow droughts, from those that could not. While the analysis was done for diverse lags, in general, the cross-correlation values and precursor coincidence rates, decreased gradually from zero lag (lag = zero months) to a lag of 12 months. Therefore, in this section, all the results are analysed only for a lag of zero months, i.e., when the SSI-1 has no time delay respect to the analysed drought indices. The remainder of the analysis can be found in the Supplementary Material (Figures S2–S4 and S5–S7).

Figure 4 displays the linear correlation values (zero lag) between the selected drought indices and the SSI-1. The highest median correlation values are obtained for SPI-6 ($\sim 0.58$), followed closely by SPI-3 and SPI-9 ($\sim 0.57$ both of them). A similar behaviour is obtained for SPEI, which yields the maximum median correlation for SPEI-3 and SPEI-6 ($\sim 0.52$). Interestingly, the values of SPEI exhibit a lower median correlation than those of the SPI, which suggests that, in the selected Chilean catchments, the variations in the climatological water balance ($P - PE$) are of less importance than the variations in $P$ alone. This happens mainly in arid regions, where the $PE$ is very high compared to the actual evaporation. Our results are in line with those of Barker et al. (2016), who found the highest correlation values between SSI-1 and SPI at temporal scales ranging from 1 to 9 over most of the 121 evaluated UK near-natural catchments; and those of Peña-Gallardo et al. (2019), who found that SPEI at temporal scales from two to four months are the most correlated with SSI-1.

The soil moisture drought index ESSMI reaches its highest median correlation value at a temporal scale of 1 month (ESSMI-1 $\sim 0.50$), followed by ESSMI-3 ($\sim 0.47$). This indicates that shorter accumulation periods of soil moisture (from the antecedent 1 to 3 months) align better with the SSI-1 signal over these catchments. Finally, the median correlation values for SWEI are all lower than 0.4 on average for all catchments, with the highest values obtained for SWEI-1 at lags from zero to five months. This suggests that, when considering all catchments collectively, snow accumulated during the cold season has a relatively small influence on the observed $Q$ in the subsequent warm season. This results indicate that nival catchments have a higher streamflow predictability as concluded by Pechlivanidis et al. (2020). In this sense, SWEI-1 in combination with other drought indices might provide useful information related to deficits and surplus of streamflow values. However, the relatively low correlation values of the SWEI could also be influenced by the ability of ERA5-Land in representing appropriately the $SWE$ over the evaluated catchments.

Figure 5 displays the spatial distribution of the linear correlation values (zero lag) between the selected drought indices and the SSI-1 at different temporal scales. The spatial patterns observed at different temporal scales are similar, therefore, here we focus on the spatial patterns obtained at zero lag and the scales that yielded the highest cross-correlation values. Figures S8–S11 in the Supplementary Material show the spatial distribution of the cross-correlation values of the selected drought indices at different temporal scales and lag values ranging from 0 to 12 months.

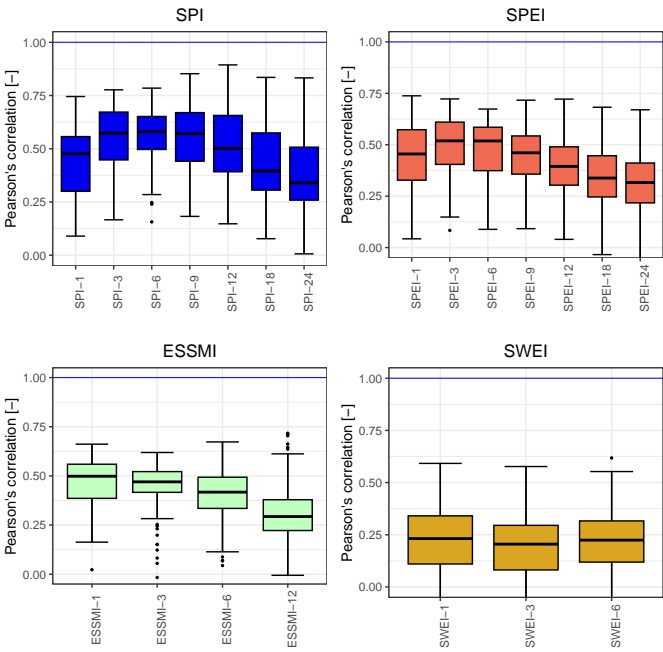

**Figure 4.** Linear correlation (zero lag) between the selected drought indices and the SSI-1 over all catchments. The horizontal blue line indicates the optimal correlation value. The solid line within each box represents the median value, the edges of the boxes represent the first and third quartiles, and the whiskers extend to the most extreme data point which is no more than 1.5 times the interquartile range from the box.

The spatial distribution of the median correlation values of SPI-6 and SPEI-3 is similar. For both indices, the highest values ($\sim$0.4–0.8) are obtained in Central and Southern Chile (30–43°S), with low values ($\sim$0.4–0.2) in the Far and Near North (17.5–30°S) and Far South (43–56.6°S, $\sim$0.4–0.0), which are related to the reduced capability of CR2METv2.5 (and other $P$ products) in representing accurately the $P$ patterns of these areas (Baez-Villanueva et al., 2020). It is worth noting that the highest values in Central Chile (32–36°S) are obtained with SPI-12 ($\sim$0.6–0.8) and SPEI-12 ($\sim$0.4–0.8), indicating that streamflow droughts are related to longer $P$ deficits over those catchments. These results are consistent to the larger hydrological memory over snow-dominated basins reported by Alvarez-Garreton et al. (2021), where streamflow is modulated by snowmelt infiltration and slow groundwater contribution to streamflow for more than a year after winter precipitation season. Large groundwater contribution in this region is coherent with the existence of important aquifers in the central valley (Taucare et al., 2024). In this same line, previous studies have reported high correlation between the SPI-12 and groundwater droughts (e.g. Bloomfield and Marchant, 2013b; Bloomfield et al., 2015; Folland et al., 2015; Apurv et al., 2017b). Furthermore, the different temporal scales suggested in Table 2 for nival and nivo-pluvial catchments agree with result obtained by Peña-Gallardo et al. (2019) for 289 catchment in the conterminous U.S. They found an important spatial heterogeneity in hydrological droughts of mountainous river basins, whose response was controlled not only by precipitation variability but also by temperature. Finally, and

even though the methodologies are not directly comparable, the temporal scales suggested for SPI and SPEI in Table 2 for all hydrological regimes but nival, partially agree with those of Meresa et al. (2023). They found that the overall lag time between the start of the meteorological and hydrological drought is almost 4–6 months, using SPI and SPEI at different temporal scales for analysing nine subcatchments of the Awash basin in Ethiopia, without distinction of the hydrological regime.

    The ESSMI-1 shows the highest median correlation values ($\sim$0.4-0.7) in Central and Southern Chile (34–43°S), low values
($\sim$0.2–0.4) in the Far and Near North (17.5–34°S) and Far South (43–56.5°S, $\sim$0.5–0.0). Similarly to SPI and SPEI, the highest values of correlation between ESSMI and SSI-1 in the Near North (32–36°S) are obtained with ESSMI-6 ($\sim$0.5–0.8). Finally, SWEI-1 shows its highest median correlation values ($\sim$0.3–0.6) in Central Chile (30–34°S) yielding relatively low values ($\leq$0.3) elsewhere.

    We used two thresholds to define a drought event in the event coincidence analysis: $i$) $-1.0$ for *moderate* drought events, and
$ii$) $-1.5$ for *severe* drought events as done in previous studies (Andreadis et al., 2005; Sheffield et al., 2009). Figure 6 shows boxplots with the precursor coincidence rates (hereafter referred to as coincidence rates) for all evaluated indices for moderate drought events ($thr = -1.0$). Coloured boxplots indicate the temporal scales at which coincidence rates are significant at a confidence level of 95% for at least 75% of the evaluated catchments (75 in this case). In contrast, white boxplots show the temporal scales where less than 75% of the catchments had statistically significant coincidence rates.

The highest median coincidence rates are obtained for SPI-6 ($\sim 0.53$), followed closely by SPI-3 ($\sim 0.52$) and SPI-9 ($\sim 0.50$). A similar behaviour is obtained for SPEI, which achieves the maximum coincidence rates for SPEI-6 ($\sim 0.51$), followed closely by SPEI-3 and SPEI-9 ($\sim 0.50$ both of them). Our results indicate that, as expected, meteorological droughts strongly control streamflow droughts and that $P$ influences meteorological droughts the most compared to the evaporative demand as suggested by the small differences found between the SPI and SPEI. On the other hand, the median coincidence rates for
ESSMI are all lower than 0.4, and they reach their highest values at a temporal scale of 3 months (ESSMI-3 $\sim 0.31$), followed by ESSMI-6 ($\sim 0.29$) and ESSMI-1 ($\sim 0.28$). Similarly to the $SWE$, the lower performance of the ESSMI could be related, to some extent, to the ability of the product in representing the soil moisture patterns across Chile.

    While the results of the cross-correlation analysis and the event coincidence analysis cannot be directly compared, it is worth noting the substantial differences between both analyses regarding the ESSMI at the evaluated lags. Although soil moisture
and $Q$ signals are related, as evidenced by the relatively high cross-correlation results between the ESSMI and the SSI-1, the substantially lower event coincidence rates between these indices suggest that a high correlation does not mean that a specific drought index could be used to assess streamflow droughts as the correlation also includes periods with near-normal and surplus conditions (in this case of soil moisture and $Q$). These differences in the case of the ESSMI could be related to the fact that soil moisture has a more immediate influence on the runoff coefficient via infiltration during rainy periods compared to its
influence on evaporation. Therefore, as correlation evaluates also periods with surplus of the evaluated variables, it provides a better agreement between the standardised soil moisture and $Q$ signals. On the other hand, our implementation of the event coincidence analysis focuses only on dry conditions making it a better method to relate the selected indices to streamflow droughts.

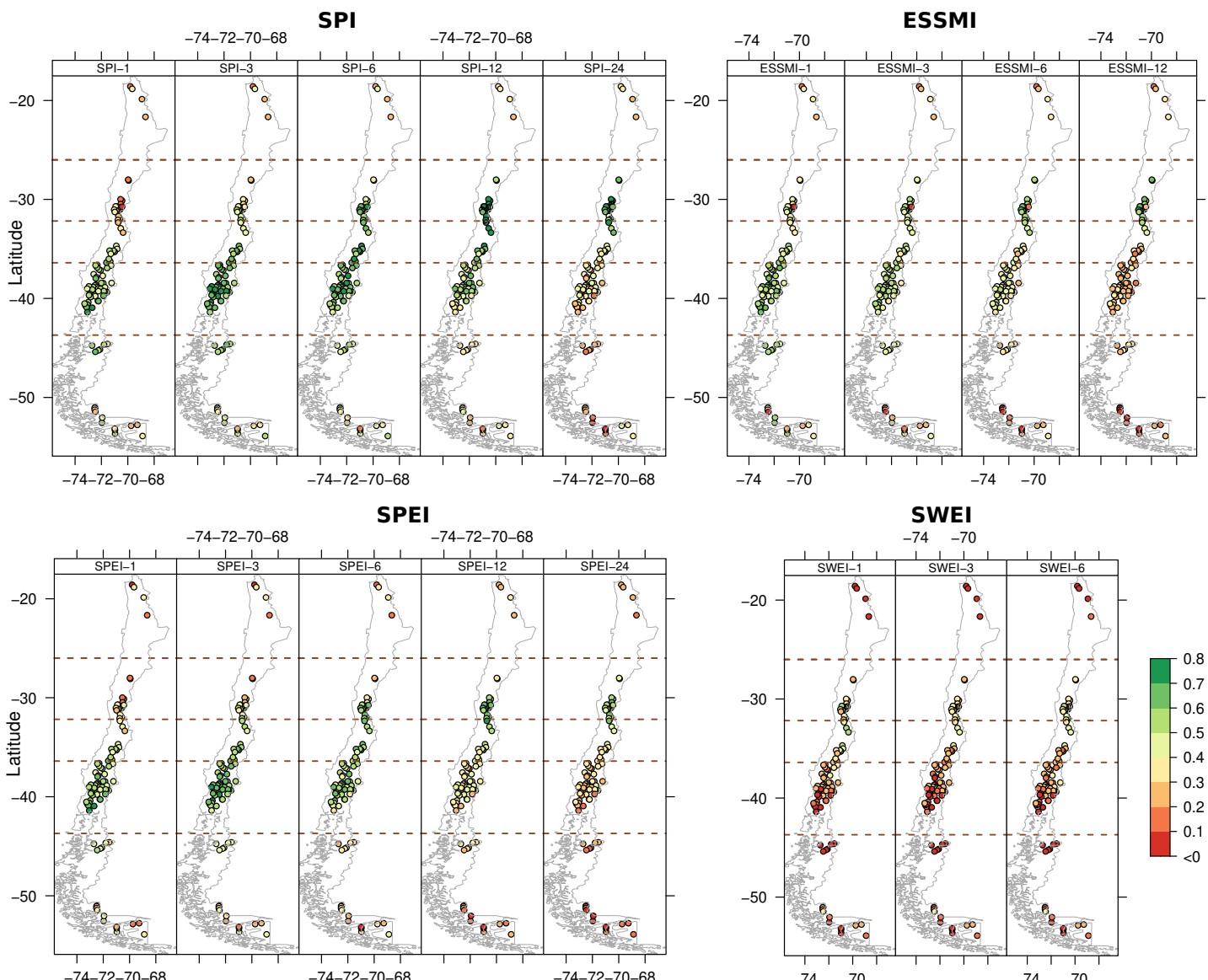

**Figure 5.** Spatial distribution of the linear correlation (lag zero) between the selected drought indices (SPI, SPEI, ESSMI, SWEI) and the SSI-1 over the 100 near-natural catchments.

The SWEI scored the lowest results (median values lower than 0.4 and not statistically significant) among the evaluated indices, which is expected since the snow influence is confined to a limited number of catchments. Finally, the median coincidence rates for SWEI are all also lower than 0.4 and not statistically significant. This can be explained by the fact that the SWEI and ESSMI have fewer events below the $-1.5$ threshold used to define severe events compared to the SSI-1, which affects the characterisation of drought events. In other words, different indices might have different thresholds related to the occurrence of

specific drought events. This is in agreement with Bachmair et al. (2015), who found that there is no single *best* threshold value
for correctly identifying the onset of drought impacts, because impacts occur within a range of threshold values. This result
suggests that the threshold used to define moderate or severe droughts for the case of the event coincidence analysis could vary
according to the component of the hydrological cycle that is being evaluated; however, the evaluation of the optimal threshold
to characterise droughts is out of the scope of this study.

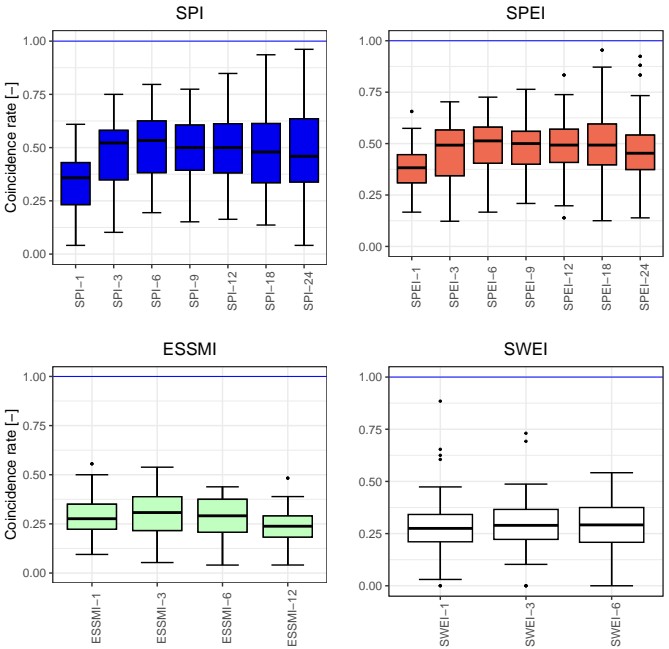

**Figure 6.** Precursor coincidence rates between the selected drought indices (SPI, SPEI, ESSMI, SWEI) and the SSI-1 at a lag of zero months
for moderate droughts ($thr = -1.0$). Coloured boxplots indicate that -at least- 75% of the catchments presented significant coincidence rates
at a 95% confidence interval, while white boxplots indicate no statistically significant rates. The horizontal blue line indicates the maximum
possible coincidence rate. The solid line within each box represents the median value, the edges of the boxes represent the first and third
quartiles, and the whiskers extend to the most extreme data point which is no more than 1.5 times the interquartile range from the box.

Figure 7 shows the spatial distribution of the coincidence rates at lag zero between the selected drought indices and the
SSI-1 at different temporal scales for moderate droughts, while Figures S12–S20 and S16–S19 in the Supplementary Material
show coincidence rates for moderate and severe droughts, respectively, considering lag values ranging from 0 to 12 months.
The spatial distribution of the median coincidence rates of SPI-6 and SPEI-6 are similar (Figure 7).

For both indices, the highest coincidence rates (∼0.4–0.8) are obtained in Central and Southern Chile (30–43°S), with
low values (∼0.0–0.3) in the Far and Near North (17.5–30°S) and Far South (43–56.6°S). It is worth noting that the highest
coincidence rates in the Near North (30–34°S) are obtained with SPI-24 (∼0.8–0.5) and SPEI-24 (∼0.8–0.3), closely followed
by SPI-12 (∼0.8–0.3) and SPEI-12 (∼0.8–0.4). This finding is consistent with Liu et al. (2019), who found weaker relationships

between the SPEI and the Standardised Runoff Index (SRI) in the arid region of the Tibetan Plateau. This could be attributed to the fact that in periods when $P$ is very low there is a point where the evaporative demand fluctuations could be deemed as irrelevant, which is consistent with periods of more than 3 to 6 months without $P$, a normal situation in northern Chile. The high accumulation periods for the SPI and SPEI (up to 24 months) is likely due to the occurrence of multi-annual droughts, which have particularly affected the central-northern region of Chile (Garreaud et al., 2020). Regarding the spatial distribution of the soil moisture index ESSMI-3, it shows its highest median coincidence rates (∼0.5–0.3) in Southern Chile (36–43°S), with low values lower than 0.4 elsewhere. Finally, SWEI-1 shows its highest median linear correlation values (∼0.6–0.3) in Central Chile (30–34°S), and lower values (≤0.3) elsewhere.

The highest cross-correlation and event coincidence rates were obtained at a lag zero, which is in agreement with Peña-Gallardo et al. (2019). Considering the results of all catchments in the cross-correlation and event coincidence analyses, it can be observed that —in general— the SPI-6, followed by SPI-3, and SPI-9 are the best indices to assess streamflow drought, with median correlation values ranging from 0.57 to 0.58, and precursory coincidence rates ranging from 0.52 to 0.53 for moderate drought events (and from 0.45 to 0.53 for severe drought events, see Table S2 of the Supplementary Material). However, in the Far and Near North, the SPI-12, SPI-24, and SPEI-12 are the best meteorological proxies of streamflow droughts, which highlights that these indices provide complementary information and that a single SPI or SPEI cannot describe the complex $P$-$Q$ interaction over Chile. Therefore, a sub-regional approach should be used considering the climatic gradient of Chile and the hydrological response of the catchments.

On the other hand, the median correlation values and coincidence rates for ESSMI at different temporal scales were all lower than 0.4, indicating that ESSMI is not a better proxy for streamflow droughts than SPI and SPEI. Similarly, Figures 4, 5, 6, and 7 show that when all the catchments are analysed at once, SWEI is not a better proxy than SPI or SPEI at any temporal scale. Overall, the results suggest that various hydrological processes are likely to influence the response of the 100 near-natural catchments situated within Chile's edapho-topo-climatologically diverse territory. Although the ESSMI and the SWEI did not yield higher values compared to the SPI and SPEI, these indices could be combined into a sequential drought index. This index could consider $P$ deficits first, followed by soil moisture deficits and/or snow deficits to mitigate the occurrence of false drought signals.

### 4.2 Influence of hydrological regime

The influence of the hydrological regime of the catchments on the relationship between different scales of meteorological, soil moisture, and snow drought indices and streamflow droughts, although expected and intuitive, is almost never considered in large-scale drought analyses and characterisations. Although evaluating all catchments together provided general recommendations for which indices could be used to assess streamflow droughts, analysing the results for catchments with similar hydrological regimes provided additional insights into the relationship between the selected drought indices and how they relate to streamflow droughts. Therefore, this section analyses the cross-correlation values and precursor coincidence rates between the selected drought indices and the SSI-1 considering the hydrological regimes of the 100 near-natural catchments:

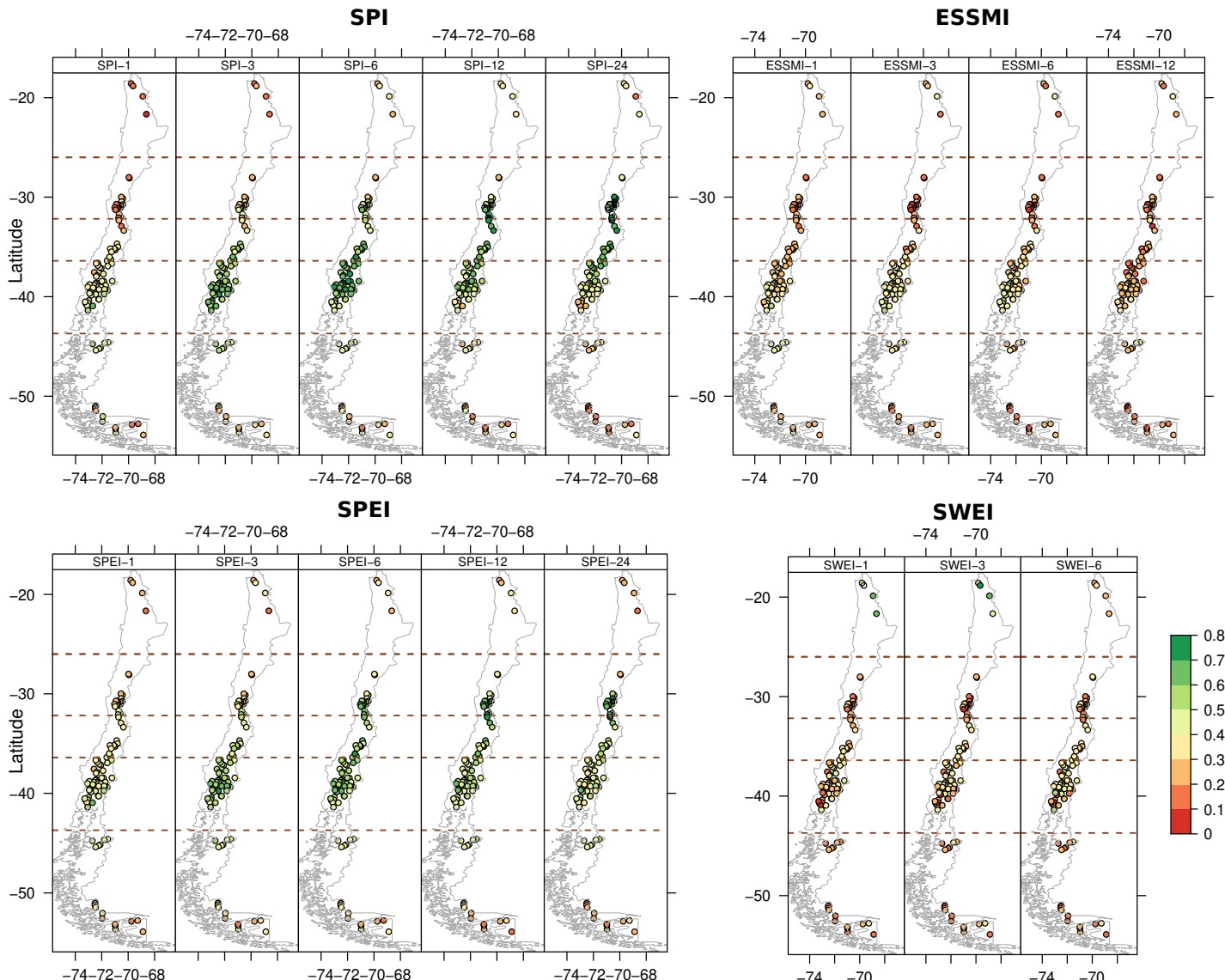

**Figure 7.** Spatial distribution of the event coincidence analysis results for moderate droughts between the selected indices and the SSI-1 over 100 near-natural catchments and at a lag of zero months.

$i$) nival (16 catchments); $ii$) nivo-pluvial (25 catchments); $iii$) pluvio-nival (40 catchments); and $iv$) pluvial (19 catchments; see Figure1).

In this sense, only cross-correlation values and precursor coincidence rates higher than 0.4 are considered as indicative that the analysed drought index and time scale is a good proxy of streamflow droughts. Figures S21–S24 and S25–S33 of the Supplementary Material show that the cross-correlation and event coincidence analysis values, in general, decreased gradually

445 from zero lag to a lag of 12 months for SPI, SPEI and ESSMI at all temporal scales, with the only exception of SPI and SPEI for nival catchments, where high correlation values and coincidence rates extend to lags of 3-6 months. Therefore, all indices are analysed hereafter only for a lag of zero months, i.e., when the SSI-1 has no delay in time with the analysed drought indices.

 Figure 8 displays the correlation values between the selected drought indices and the SSI-1 at a lag of zero months. For nival catchments the highest median correlation values are obtained for SPI-12 ($\sim 0.75$), followed by SPEI-12 ($\sim 0.51$), ESSMI-6 ($\sim$

450 $0.51$) and SWEI-6 ($\sim 0.51$). Nivo-pluvial catchments show a slightly different behaviour, with the highest median correlation values obtained for SPI-12 ($\sim 0.52$), followed by ESSMI-3 ($\sim 0.49$), SPEI-3 ($\sim 0.48$), and SWEI-1 being a poor proxy of streamflow drought ($\sim 0.38$). The highest median correlation values for pluvio-nival catchments are obtained for SPI-3 ($\sim$ $0.73$), followed by SPEI-3 ($\sim 0.68$), ESSMI-3 ($\sim 0.55$) and SWEI-1 ($\sim 0.25$). Finally, for pluvial catchments the highest median correlation values are obtained for SPEI-3 ($\sim 0.63$), followed by SPI-3 ($\sim 0.60$), ESSMI-1 ($\sim 0.52$) and, as expected,

455 SWEI-3 not being a good proxy of streamflow drought ($\sim 0.1$). While the SPI, SPEI, and ESSMI generally exhibited higher values across all regimes, the SWEI showed relatively high values in nival catchments (see Fig. S24 of the Supplementary Material) for lags from 3 to 6 months (75th percentile $< 0.60$ and median values $\geq 0.50$). These values are higher compared to the $P$-based indices at the same temporal scales, indicating that snow accumulation and snowmelt processes are important drivers of $Q$. This was followed by nivo-pluvial (75th percentile $< 0.49$) and pluvio-nival (75th percentile $< 0.28$) catchments.

460 These findings are consistent with Tijdeman et al. (2018), who obtained high correlation values over cold climates in the US, and with Staudinger et al. (2014), who found similar results over alpine catchments in Switzerland. The larger scales of the SPI and SPEI over nival catchments are related to the fact that $Q$ is mostly controlled by snow melt that occurs after snow accumulation; contrastingly, the pluvial catchments react rapidly to $P$ and therefore, $P$ deficits are more quickly propagated into $Q$ deficits.

465 The larger time lags observed for SPI and SPEI over snow-dominated basins, in contrast to the shorter lags in pluvial basins, along with the increased lag for SWEI, are consistent with how catchment memory modulates the propagation of solid and liquid precipitation into the hydrological cycle. This, in turn, determines the time until a streamflow drought is influenced by rainfall and snow-related processes (Alvarez-Garreton et al., 2021).

 Figure 9 shows the coincidence rates between the selected drought indices and the SSI-1 for moderate drought events

470 ($thr = -1$) with zero lag. For all hydrological regimes, ESSMI and SWEI presented median coincidence rates lower than 0.4 (i.e., less than 40% of streamflow droughts preceded by soil moisture and snow droughts) for all time scales and, therefore, are not considered good proxies of streamflow drought events. For nival catchments, the highest coincidence rates are obtained for SPI-24 ($\sim 0.74$), followed by SPEI-18 ($\sim 0.52$). Nivo-pluvial catchments show a different behaviour, with the highest median correlation values obtained for SPEI-9 ($\sim 0.50$), followed by SPI-6 ($\sim 0.48$). The highest median correlation values for pluvio-

475 nival catchments are obtained for SPI-6 ($\sim 0.68$), followed by SPEI-3 ($\sim 0.60$). Finally, for pluvial catchments the highest median correlation values are obtained for SPI-3 ($\sim 0.55$), followed by SPEI-3 ($\sim 0.49$). The results of the event coincidence analysis per hydrological regime for severe drought events ($thr = -1.5$) show a behaviour similar to that of moderate droughts, but with lower values, except for SPI and SPEI in pluvio-nival catchments, where coincidence rates for severe events are higher than those of moderate ones (Figure S25).

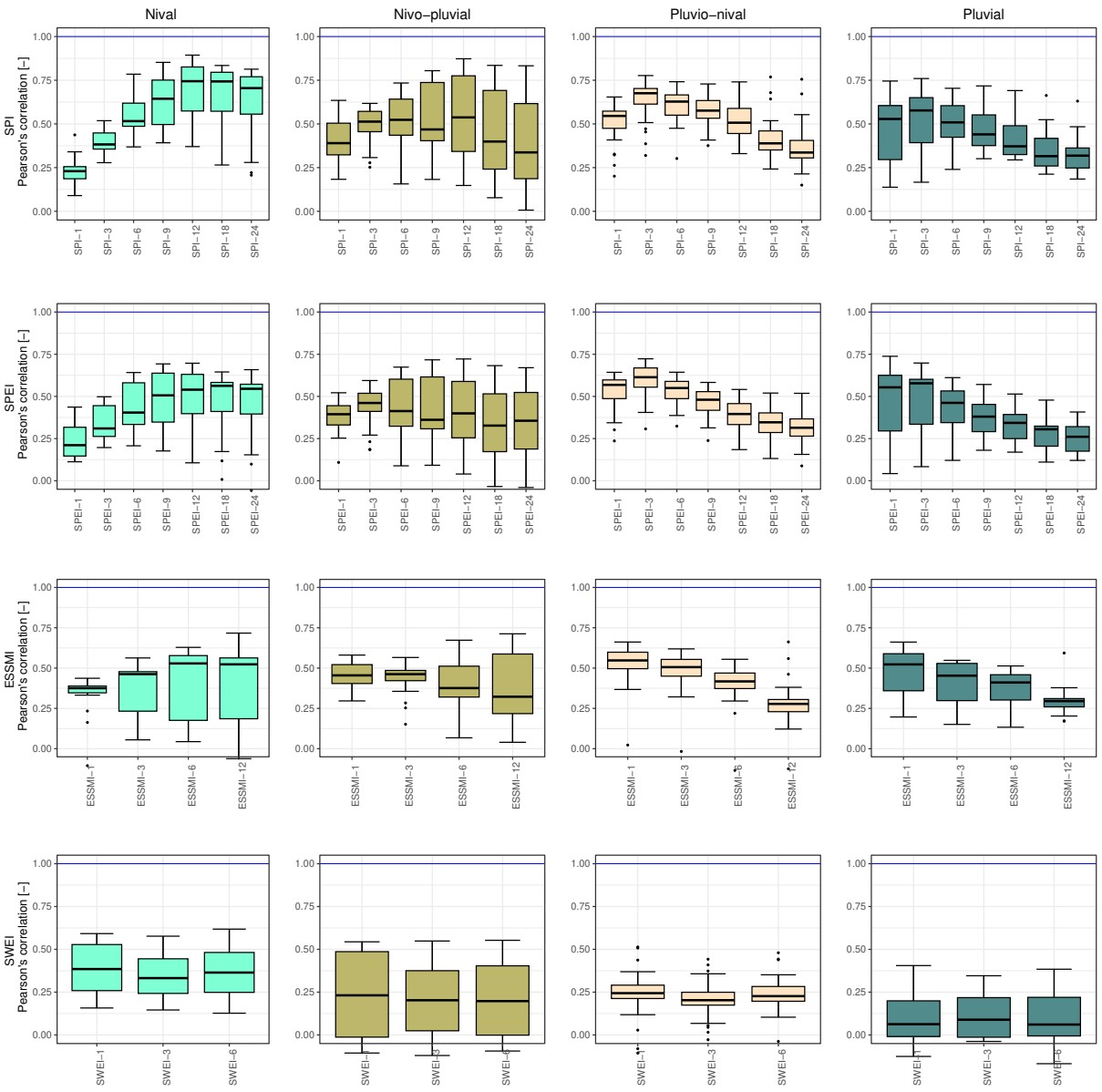

**Figure 8.** Linear correlation values between the selected drought indices (SPI, SPEI, ESSMI, SWEI) and the SSI-1, using a lag of zero months for SPI, SPEI and ESSMI and a lag of 6 months for SWEI. From left to right, the four columns corresponds to $i$) nival; $ii$) nivo-pluvial; $iii$) pluvio-nival; and $iv$) pluvial catchments. The horizontal blue line indicates the optimal cross-correlation. The solid line within each boxplot represents the median value, the edges of the boxes represent the first and third quartiles, and the whiskers extend to the most extreme data point which is no more than 1.5 times the interquartile range from the box. A detailed summary of the highest results can be found in Table S3 of the Supplementary Material.

Our results show that the SPI scored the highest cross-correlation and event coincidence values among all evaluated indices, which can be attributed to the fact that $P$ is the most important variable that drives the $Q$ response. Similarly, Barker et al. (2016) found high cross-correlation values between the SPI-1 and SPI-3, and the SSI-1 over the majority of 121 near natural catchments in the UK. The event coincidence analysis showed relatively low values of the ESSMI compared to those obtained with the cross-correlation analysis, indicating that even if an index has a high correlation with the SSI-1, it may not be able to accurately detect specific streamflow drought events. This result highlights the fact that a high correlation between two indices does not imply that they can be interchangeably used to assess drought conditions. The SWEI showed relatively high values in pluvio-nival catchments, which could be related to $T$ increases (Garreaud et al., 2020) and the loss of snow cover over these catchments (Notarnicola, 2022). However, these values were substantially low compared of those of the cross-correlation analysis, which could be attributed to the fact that nival catchments present 5-6 months of delay between the rainfall season (June–August) and the months with maximum discharges (October–December), reinforcing the idea that a correlation analysis does not necessarily indicate that and index could be used to assess streamflow droughts.

Additionally, these results suggest that the SWEI alone is not capable of representing $Q$ droughts in snow-influenced catchments, which could be due to ERA5-Land's limitations in representing $SWE$ in these areas. However, combining it with other indices might be beneficial, which is in line with previous studies highlighting the importance of incorporating snow-related information in drought analysis and prediction (Huning and AghaKouchak, 2020; Gottlieb and Mankin, 2021).

Figure 10 depicts the spatial distribution of the temporal scale with the highest correlation (zero lag) and precursor coincidence rates for all indices. These results show that, although the cross-correlation analysis and event coincidence analysis are in close agreement concerning the spatial distribution of the recommended scales for each index, the event coincidence analysis suggests slightly higher scales compared to the cross-correlation analysis. For example, over the South region, the SPI provides the highest results between accumulation periods of 1–9 months, while for the event coincidence analysis for moderate droughts, scales up to 12 months are visible.

The drought indices are presented in Table 2. These drought indices were chosen to accommodate the diversity of the catchments. These results suggest that the greater the snow influence in the catchments, the higher the accumulation period required for the variables in the evaluated meteorological, soil moisture, and snow drought indices. These results are in line with Fluixá-Sanmartín et al. (2018), who recommended the use of SPI-6 to assess droughts using a catalogue of historical droughts over the snow-influenced Jinsha River Basin in China, and with Vicente-Serrano and López-Moreno (2005) who determined that $Q$ respond to short SPI time scales in the snow-influenced high basin of the Aragon River in Spain.

This evaluation provides a new perspective on the importance of the hydrological regime of catchments in the relationship between meteorological, soil moisture, and snow indices and streamflow droughts, with specific focus on the complex topography and climate of Chile. Previous studies have focused primarily on the influence of geomorphological catchment characteristics on these relationships. For example, Peña-Gallardo et al. (2019) found that elevation is an important driver in the response of SSI to SPEI, while Van Loon and Laaha (2015) concluded that drought deficit over Austrian catchments is influenced by catchment average wetness and elevation. Similarly, our findings show that nival catchments, which are located at higher elevations, exhibit different behaviour in the response of the evaluated indices to SSI-1 compared to catchments with

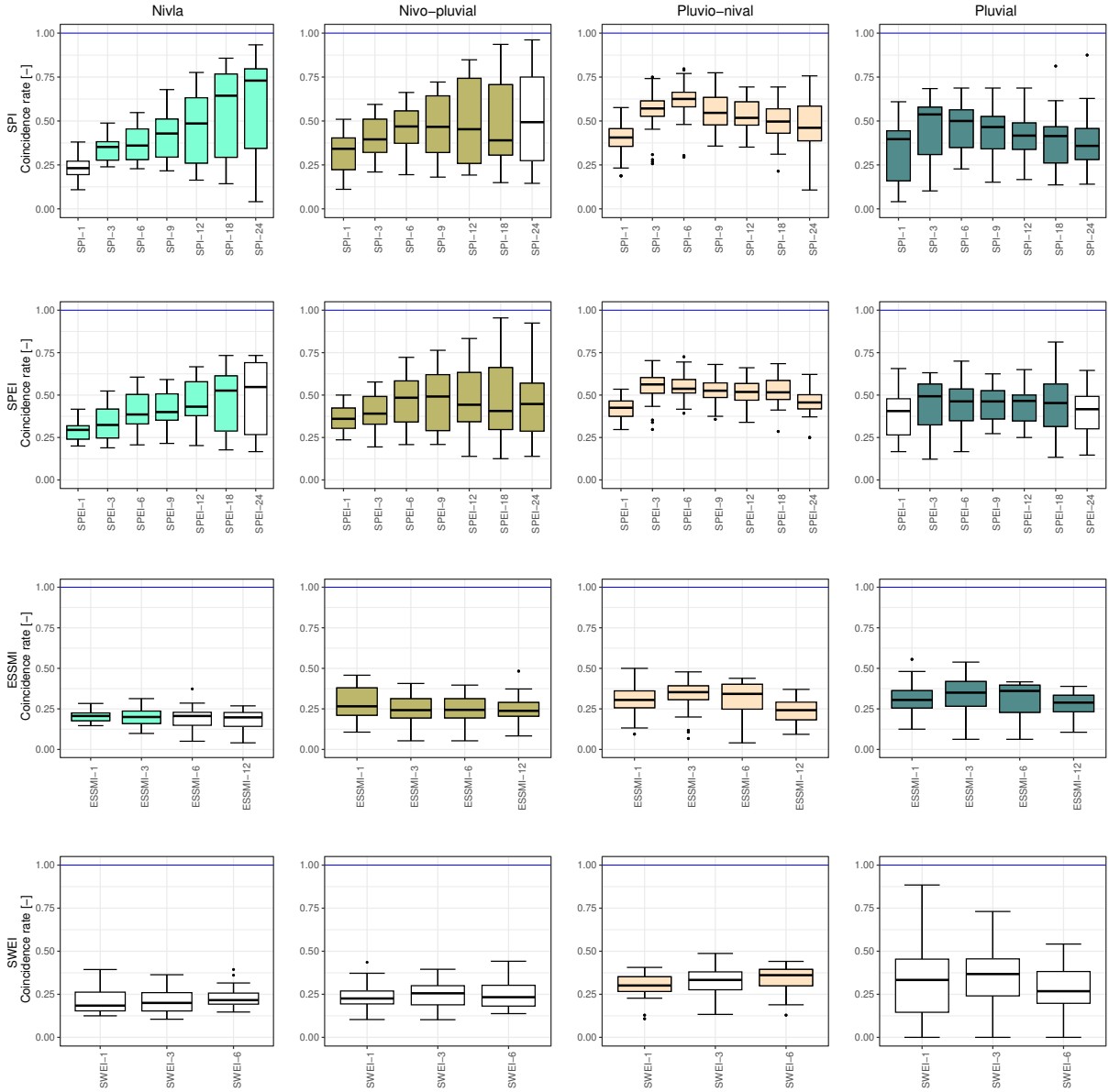

**Figure 9.** Event coincidence analysis results of the selected drought indices and the SSI-1 for moderate droughts at a lag of zero months for $i$) nival; $ii$) nivo-pluvial; $iii$) pluvio-nival; and $iv$) pluvial catchments. Coloured boxplots indicate lags where at least 75% of the catchments presented significant results at the 95% confidence interval, while white boxplots indicate the opposite. The blue line indicates the optimal cross-correlation. The solid line within each box represents the median value, the edges of the boxes represent the first and third quartiles, and the whiskers extend to the most extreme data point which is no more than 1.5 times the interquartile range from the box. A detailed summary of the highest results can be found in Table S4 of the Supplementary Material.

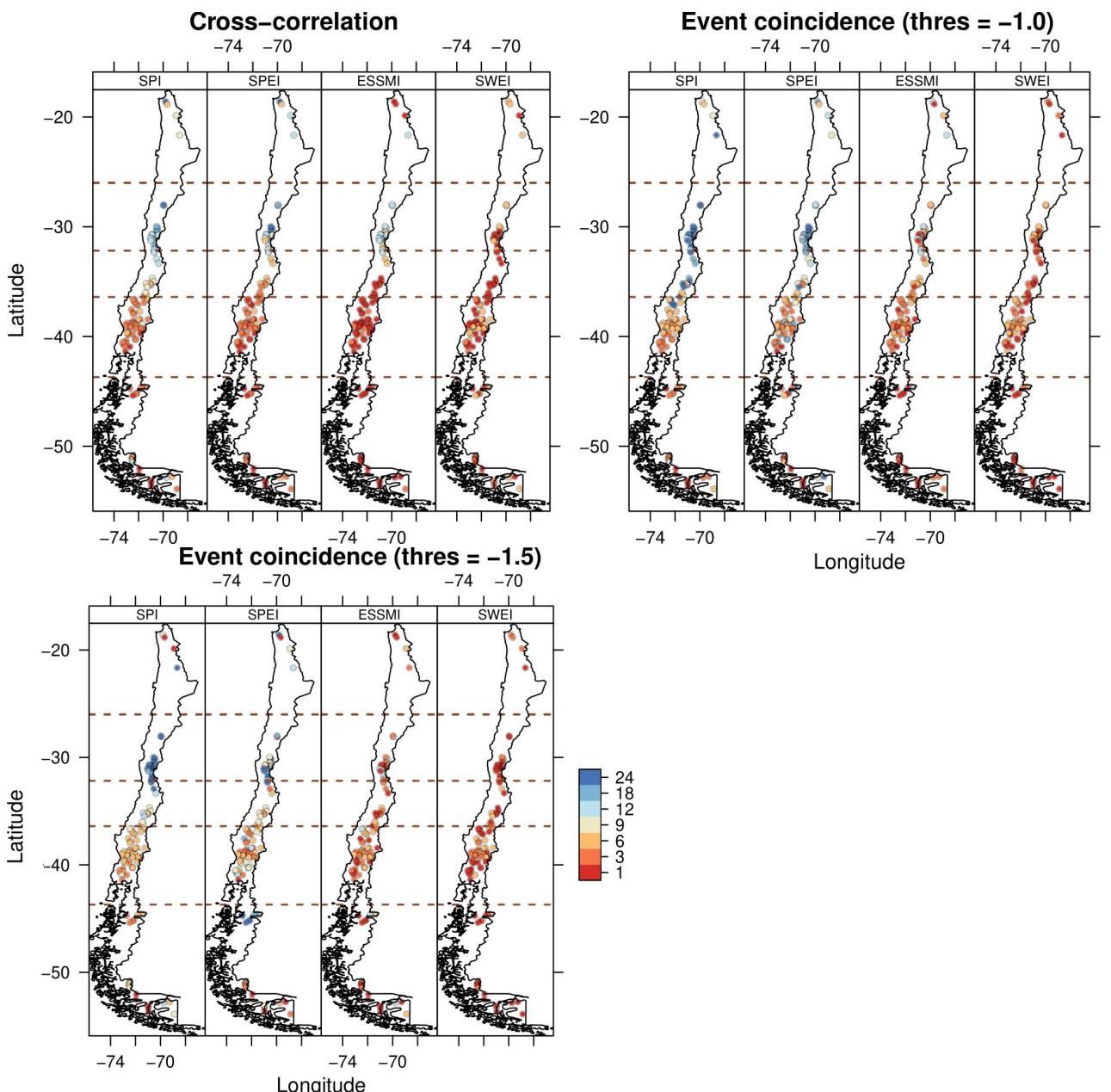

**Figure 10.** Spatial distribution of the scale that scored the highest correlation (left panel) and precursor coincidence rates (middle and right panel for moderate and severe droughts, respectively) at lag zero over 100 near-natural catchments for each index.

other hydrological regimes. Additionally, it is worth also highlighting the different behaviour of central-northern catchments that exhibit longer accumulation periods compared to the more southern, wetter catchments.

**Table 2.** Drought indices proposed to assess streamflow droughts according to the hydrological regime of a catchment.

| Hydrological Regime | Scales | | | |
|:---:|:---:|:---:|:---:|:---:|
| | SPI | SPEI | ESSMI | SWEI |
| Nival | 12–24 | 12–18 | 6–12 | - |
| Nivo-pluvial | 3–12 | 3–6 | 1–3 | - |
| Pluvio-nival | 3–6 | 3–6 | 1–3 | - |
| Pluvial | 3–6 | 1–6 | 1–3 | - |

## 5  Conclusion

Assessing streamflow droughts in data-scarce regions can be challenging due to the unavailability of $Q$ data (Peña-Gallardo et al., 2019), which can cause mismatches between the streamflow drought hazards and their related impacts. Publicly available gridded datasets of $P$, air temperature, evaporation, soil moisture, and $SWE$, among others, can be used to compute drought indices in data-scarce areas and serve as a proxy to assess streamflow droughts. In this study, the CR2MET Chilean dataset was used to provide $P$ and $PE$ data, while soil moisture and $SWE$ were collected from ERA5-Land for 1979–2020. We used 100 near-natural catchments in Chile to evaluate which drought indices and temporal scales can be used as proxies of streamflow droughts in catchments with four main types of hydrological regimes (nival, nivo-pluvial, pluvio-nival, pluvial). First, we computed the SPI, SPEI, ESSMI, and SWEI at different temporal scales in the selected catchments. Second, we applied a cross-correlation and event coincidence analysis between the previous indices and the SSI-1, as representative of streamflow drought events. Finally, linear correlation and precursor coincidence rates were analysed for all catchments simultaneously and separated by the hydrological regime. Our results indicate that there is no single best index and temporal scale that could be used as a proxy for streamflow droughts, and that this relationship depends, to some extent, on the hydrological regime of the catchments, and therefore, on their hydrological behaviour. In the following lines we summarise our key findings:

1. There is no single meteorological, soil moisture, or snow drought index and temporal scale that could be used to characterise streamflow droughts in the diverse climate regions of Chile. The best index and temporal scale depend on the hydrological regime of each catchment.

2. Snow-dominated catchments have a larger memory and consequently, larger temporal scales of the drought indices can be used as proxy of streamflow drought. In general terms, snow accumulation and snowmelt processes are important drivers of $Q$. This influence is evident in the relatively high cross-correlations observed between the SSI and the snow accumulation index lagged 3–6 months or the $P$-based indices accumulated over 6 months.

3. The nival catchments show a less pronounced decrease in cross-correlation and event coincidence values at increasing lags, and therefore, the evaluated indices can be used to assess streamflow droughts up to five or six months in advance. Contrastingly, shorter memory pluvio-nival and pluvial catchments show a rapid decay in cross-correlation and event coincidence values at increasing lags, and therefore low predictability of streamflow droughts.

4. For the Chilean case study in particular, the following drought indices are advised (individually or jointly) to monitor streamflow droughts depending on the hydrological regime of the study area: $i$) for nival catchments: SPI-12–24, SPEI-18; $ii$) for nivo-pluvial catchments: SPI-6–12, SPEI-3–9; $iii$) for pluvio-nival catchments: SPI-3–6, SPEI-3; and $iv$) for pluvial catchments: SPI-3 and SPEI-3.

5. Notwithstanding soil moisture is a key variable modulating the propagation of meteorological deficits into hydrological ones, meteorological indices are better proxies of streamflow droughts than soil moisture ones.

6. Interestingly, the cross-correlation values and event coincidence rates of the SPI were generally higher than those of the SPEI, indicating that for the selected Chilean catchments, the variations in the climatological water balance ($P - PE$) are of less importance than the variations in $P$ alone. This was more pronounced over arid regions, where the $PE$ is very high compared to the actual evaporation.

7. As the cross-correlation analysis includes the full signal of the evaluated indices and the SSI-1 (periods with surplus and deficits), a high cross-correlation value does not necessarily imply that the evaluated drought indices could be used to assess streamflow drought conditions. Therefore, future studies evaluating the use of meteorological, soil moisture, and snow drought indices as proxies of streamflow drought should consider methods that evaluate solely drought conditions, such as the event coincidence analysis.

Mountainous catchments play a pivotal role in supplying freshwater resources to both society and ecosystems. They contribute substantial runoff, redistribute winter $P$ to spring and summer, and mitigate flow variability in lowlands (Viviroli and Weingartner, 2004; Viviroli et al., 2011). Numerous studies have demonstrated the impact of temperature increases and changes in snowfall fraction on droughts in mountainous areas (Marke et al., 2018; Blahušiaková et al., 2020), potentially affecting both the total volume of streamflow and its seasonality. Given that mountainous catchments generally remain ungauged, it is crucial to identify which drought indicators and timescales of monitored variables can be used to assess streamflow drought conditions. The approach demonstrated in this paper highlights the effectiveness of using individual drought indices to characterise streamflow drought, while also highlighting the varying behaviour of these indices across catchments with diverse hydrological regimes. This suggests that combining different drought indices based on catchment-specific responses (e.g., catchment memory Alvarez-Garreton et al., 2021) could provide further avenues for strengthening the relationship between meteorological, soil moisture, and snow drought indices as proxies of streamflow drought. Although catchment characteristics are important in understanding the relationship between meteorological, soil moisture, snow drought and streamflow droughts, this article provides a complementary perspective on how this relationship can also be related to the hydrological regime of the catchments. Our findings indicate that hydrological regimes offer valuable insights into the time lag at which these indices could be used to assess streamflow drought.

In general terms, the cross-correlation and event coincidence analysis values decreased gradually from a lag of zero months to a lag of 12 months. This could be explained by the fact that only processes that cause a strong temporal shift in response (like snow accumulation and snowmelt) will show a clear lag response. An optimal lag of zero months was found by (Peña-Gallardo

et al., 2019), which indicates that these shifts generally do not occur, even in slowly-responding catchments. Specifically, nival catchments exhibited relatively high rates of event coincidence at lags of up to 5 months in the case of SPI and SPEI. These results suggest that current SPI and SPEI values could hold predictive value to assess future streamflow droughts. In contrast, the rapid decline in coincidence rates at increasing lags over pluvial catchments makes it challenging to employ these indices for projecting future streamflow droughts over these catchments. The relatively low values achieved by the ESSMI and SWEI may, in part, be attributed to the ability of ERA5-Land in capturing soil moisture and $SWE$ patterns, along with its relatively coarse spatial resolution concerning the size of the Chilean catchments under examination. The SWEI standardises the time series of $SWE$, indicating a lack of snow accumulation. The low correlation values and event coincidence rates observed in the evaluation of the SWEI may be also attributed to the fact that the SWEI does not explicitly account for snowmelt and its impact on streamflow conditions. The effects of low snow accumulation become noticeable in streamflow only during the melt season; hence, lower correlations between the SWEI and SSI are expected. For future studies aiming to assess the influence of snowmelt on streamflow drought, we recommend using the Standardised Snow Melt and Rain Index (SMRI; Staudinger et al., 2014) instead of the SWEI. The SMRI requires the implementation and calibration of a snow model capable of representing snow processes in the target catchments.

Additionally, we used both parametric and non-parametric approaches to compute the drought indices, aligning with the methods established in their respective literature. However, the selection between parametric and non-parametric approaches can impact the computed values of the respective indices (Soláková et al., 2014; Mallenahalli, 2020) and their associated drought characteristics (Tijdeman et al., 2020). To illustrate this, Figure S34 of the Supplementary Material shows the differences of a standardised soil moisture index using both a parametric approach with the gamma distribution and the non-parametric approach proposed by (ESSMI Carrão et al., 2016) focusing on a nival and pluvial catchment. The combination of parametric and non-parametric approaches to classify drought events can potentially increase the uncertainty in the evaluation, as the distribution of standardised values may differ among indicators. The selection of parametric or non-parametric approaches could impact the occurrence of extreme events and, consequently, the correlation values and event coincidence rates. For instance, in the case of the non-parametric ESSMI and SWEI, less extreme values were observed compared to the SPI and SPEI parametric approaches.

Future studies could focus on $i$) evaluating the uncertainties related to the selection of parametric or non-parametric approaches for calculating diverse drought indices across multiple temporal scales; $ii$) analysing the relative changes in performance of these indices using adjusted soil moisture and $SWE$ datasets; $iii$) evaluating the sensitivity of the selected threshold to classify droughts; and $iv$) characterising the seasonality, total duration, severity, and intensity of catchments with different hydrological regimes. We hope this information would help to improve current drought monitoring and shift towards more effective and proactive management strategies.

*Author contributions.* OMBV and MZB conceptualised the research. OMBV, MZB, and DGM wrote the manuscript with help of all authors. OMBV designed the computational framework and analysed the data. DGM and JFS verified the analytical methods. DGM, HEB, JFS, CAG, KV, RG, and MG contributed to the interpretation of the results. All authors provided critical feedback and helped shape the research, analysis, and manuscript.

*Competing interests.* No competing interests

*Acknowledgements.* We extend our sincere gratitude to Anne Van Loon for efficiently handling the manuscript and for her valuable insights and recommendations. Additionally, we appreciate the contributions of Juan Diego Giraldo-Osorio and the two anonymous reviewers, whose constructive comments significantly enhanced the quality of the final manuscript. Special thanks to the HESS editorial team for their support, and Rodrigo Marinao Rivas for his help with sections 8 and 9 of the Supplementary Material. Lastly, the authors acknowledge the
support from the following projects: ANID-Sequía FSEQ210001, ANID Fondecyt Regular 1212071, ANID-PCI 190018, and ANID/FON-DAP/1522A0001.

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
