# Peer review of "On the timescale of drought indices for monitoring streamflow drought considering catchment hydrological regimes"

_EGUsphere, 2023_

## Referee Comment (RC1)

**1. Title**

On the time scale of meteorological, soil moisture, and snow drought indices to assess streamflow drought over catchments with different hydrological regime: a case study using a hundred Chilean catchments

**2. Research question**

From abstract and introduction

- i) what temporal scales of (the frequently used and easy to calculate) SPI and SPEI meteorological indices can be used as proxies for streamflow drought in catchments with different hydrological regimes?
- ii) considering that the soil acts as a natural reservoir to maintain streamflow during periods of reduced P, can a soil moisture drought index be used to assess streamflow droughts instead of the SPI and SPEI meteorological indices?
- iii) considering that snowmelt is an important moisture contribution to streamflow and surface water availability during spring and summer months in catchments with a pronounced snow influence, can a SWE (Snow Water Equivalent) index be used to assess streamflow droughts instead of the SPI and SPEI in those catchments?

**3. Goals**

From introduction and research question

- To define temporal scales for SPI and SPEI for using them as proxies for streamflow droughts in different hydrological regimes.
- To evaluate the suitability of a soil moisture drought index for using it instead of SPI and/or SPEI meteorological indices to assess streamflow drought.
- To evaluate the suitability of SWE index to assess the streamflow droughts in snow-influenced catchments.

**4. Findings**

From abstract

- i) there is no single meteorological, soil moisture, or snow drought index and temporal scale that could be used to characterise all streamflow droughts across Chile, and
- ii) the greater the snow influence in a catchment, the larger the temporal scale of the drought index to be used as proxy of streamflow drought.

**5. How do you rate this paper in absolute terms? Poor to fair, good, very good to excellent.**

Good to very good.

**6. Recommendation.**

My decision is between "accepted after minor revisions" to "accept without changes".

**7. Confidential comments to the editor**

The paper has been well written: the results are clear (however, some improvements are still possible), well presented, and explained. The research questions are pertinent because it is necessary to improve the understanding of drought propagation in the hydrological cycle.

The proposed minor changes in the document (in "comments to the author") do not affect the paper quality. Also, I made some specific questions that must be easily answered by the authors.

That is why my decision between "accepting" and "accepting after minor changes".

**8. Comments to the author**

I had a hard time following the explanation of the results of the Spearman correlation analysis. What variables were compared in this analysis? I believe it is easier to present these results in a table, preserving the text explanation. The text alone makes it tremendously difficult to understand what is meant.

**9. Questions**

How do you feel about your assumption that the SSI-1 is the most representative of streamflow deficit conditions for the several catchments? Is it possible that catchments with higher storage (e.g., snow-influenced basins) should characterize their hydrological droughts with SSS for longer durations? Why yes, or why not, do you believe that SSI-1 is suitable for all your catchments?

You say that SSI-1 integrates catchment-scale hydrological processes. I think that it would be feasible for Chilean catchments ¿Is it (the SSI-1) suitable for larger catchments than those catchments in Chile? ¿Do you think that SSI-1 integrates the catchment-scale processes in larger basins?

Is the SWE suitable for all snow basins in Chile??? If the answer is no, ¿is that the reason why the spatial scales 1, 3, and 6 months were selected?

Is it possible to use other linear or non-linear analyses to establish the relationship between the indices and SSI-1? Correlation analysis gives a first look at the relationships, but it is a linear version of the analyses. It may be that non-linear relationships that are possibly hidden in the treated data cannot be seen. Here a question arises: is the Event Coincidence Analysis (ECA) strong enough as an alternative method of assessing the relationship between droughts and their triggers? ¿Why?

In the case of the ECA, ¿were the same thresholds selected for the several indices? That is, if moderate hydrological drought events were selected in the SSI-1 (SSI-1<-1.0), it was assumed that they were triggered by moderate meteorological/soil/snow events (e.g. SPI<-1.0, SPEI<-1.0, ...). ¿Do you consider merging the severities of hydrological and meteorological/soil/snow droughts? (e.g. that some severe droughts SSI-1<-1.5 are triggered by moderate meteorological droughts: -SPI<-1.0, or SPEI<-1.0-).

Pag. 13, you say:

*While the analysis was done for diverse lags, the cross-correlation values and precursor coincidence rates, decreased gradually from zero lag (lag = zero months) to a lag of 12 months.*

¿Is true that sentence for all the analyzed catchments? If the answer is "yes", ¿what do you think is the process that drive the flow in the catchments? The response is especially interesting in the nival and nivo-pluvial catchments, where the runoff should be delayed some weeks with respect to snow precipitation.

---

## Author Comment (AC1)

**Revision of HESS-2023-1911**

Oscar M. Baez-Villanueva, Mauricio Zambrano-Bigiarini, Diego G. Miralles, Hylke E. Beck, Jonatan F. Siegmund, Camila Alvarez-Garreton, Koen Verbist, René Garreaud, Juan Pablo Boisier, Mauricio Galleguillos

**Reviewer 1**

**R1–C1: I had a hard time following the explanation of the results of the Spearman correlation analysis. What variables were compared in this analysis? I believe it is easier to present these results in a table, preserving the text explanation. The text alone makes it tremendously difficult to understand what is meant.**

We extend our sincere gratitude to Juan Diego Giraldo-Osorio (Reviewer 1) for the insightful comments and constructive feedback provided. We firmly believe that the questions and suggestions raised during the review process have significantly contributed to improving the overall quality of the manuscript.

In response to the initial comment, we agree that the explanation of the Spearman correlation analysis was unclear and did not contribute significantly to our results. Consequently, we have opted to omit this analysis from the revised manuscript to enhance overall readability.

**R1–C2: How do you feel about your assumption that the SSI-1 is the most representative of streamflow deficit conditions for the several catchments? Is it possible that catchments with higher storage (e.g., snow-influenced basins) should characterise their hydrological droughts with SSS for longer durations? Why yes, or why not, do you believe that SSI-1 is suitable for all your catchments?**

We greatly appreciate your comment (and the one of Reviewer 2 [R2-C2]). We understand the concern about our assumption regarding the SSI-1 drought index as representative of streamflow deficits. We would like to address your comments regarding our choice of the SSI-1 by complementing the information already presented in Lines 189-192:

1. While the SSI can theoretically be calculated for any accumulation period (SSI-$n$), we want to emphasise that the streamflow at the current month already integrates catchment-scale hydrometeorological and hydrogeological processes that have occurred in previous months (Stahl et al., 2020). Therefore, by using the SSI-1 we aim to assess how deficits of precipitation, soil moisture and snowpack propagate to streamflow deficits. The choice of the SSI-1 allows a direct comparison between the streamflows at the current month and the anomalies of the selected variables of the hydrological cycle ($P$, $P-PE$, $SM$, and $SWE$) accumulated at different temporal scales. This comparison provides insights into the time required for different hydrological processes to manifest as streamflow deficits (Barker et al., 2016) considering also the hydrological regime of the selected catchments.

2. Moreover, to explicitly consider the delayed impact on $Q$ that the variations in the selected variables might have, we also considered lags from 0 to 12 months for SPI, SPEI and ESSMI, and lags from 0 to five months for SWEI.

3. While we acknowledge that SSI-1 is susceptible to short-term fluctuations in streamflows (albeit to a lesser extent than SPI-1 or SPEI-1, as shown in Figure 1), our rationale in using SSI-1 is related to our focus on evaluating the impact of relatively short-term fluctuations of $P$, $P-PE$, $SM$, and $SWE$ on streamflows (1–3 months). We consider that aggregating previous months for the case of the SSI will indeed smooth the signal but at the expense of losing the overview of the actual streamflow conditions. Therefore, by accumulating streamflow deficits at temporal scales larger than 1 month, we would loose the comparability of the selected indices at shorter time scales with streamflows deficits at longer time scales (e.g., the comparison between the SPI-1 or SPI-3, against the SSI-6 has little physical meaning as the accumulation of streamflow of six months is influenced by 6 months of $P$ or more).

4. Finally, the selection of the Standardised Streamflow Index with a one-month accumulation period (SSI-1) is based on both the aforementioned arguments and its widely use in the scientific literature (e.g., Vicente-Serrano et al., 2012; Barker et al., 2016; Huang et al., 2017; Peña-Gallardo et al., 2019b; Stahl et al., 2020; Tijdeman et al., 2020b; Li et al., 2020; Wang et al., 2020; Bevacqua et al., 2021; Wu et al., 2022).

[Figure]

**Figure 1.** Time series of the SSI-1 and SPI/SPEI indices at different accumulation periods (1, 3, 6, and 12 months) for the nival Estero Yerba Loca catchment (X05721001).

To clarify this point, the following changes were made in the manuscript:

1. We expanded the reasoning of the use of SSI-1 between lines 230–236 and now reads: *Here, we used the SSI with a one-month accumulation period (SSI-1) to characterise streamflow drought because: i) it integrates catchment-scale hydrogeological processes (Barker et al., 2016) and the hydrometeorology (Stahl et al., 2020) from the preceding months; ii) while the SSI-1 may be susceptible to short-term fluctuations (to a lesser extent than SPI-1 and SPEI-1), its selection facilitates a direct comparison between the current month's streamflows and the accumulated values of selected hydro-*

*logical cycle variables (P, P − PE, SM, and SWE) at various temporal scales; and iii) it allows for contrasting with*
50 *other studies that used the same index for streamflow drought assessments (Barker et al., 2016; Li et al., 2020; Stahl et al., 2020; Wang et al., 2020; Bevacqua et al., 2021);*

2. we included a sentence in line 232 emphasising that the SSI-1 may be susceptible to short-term fluctuations in streamflows; and

3. we expanded the explanation regarding the selection of the catchments (please refer to R2-C8).

55 **R1–C3: You say that SSI-1 integrates catchment-scale hydrological processes. I think that it would be feasible for Chilean catchments ¿Is it (the SSI-1) suitable for larger catchments than those catchments in Chile? ¿Do you think that SSI-1 integrates the catchment-scale processes in larger basins?**
We appreciate the comment from Reviewer 1 and kindly direct him to our response to comment R1–C2 for further clarification.

60 **R1–C4: Is the SWE suitable for all snow basins in Chile??? If the answer is no, ¿is that the reason why the spatial scales 1, 3, and 6 months were selected?**
The rationale behind incorporating the Snow Water Equivalent Index (SWEI) was to account for snowmelt processes within the selected catchments, as they constitute a significant contribution to streamflow and surface water availability during the spring and summer months in catchments with a more pronounced influence of snow.

65 In the previous version of the manuscript it was not clear that the SWEI is not suitable for all the analysed nival catchments. In our revised manuscript we emphasise more clearly that the SWEI struggles in representing the surpluses and deficits of streamflows over snow-influenced catchments, as mentioned in lines 481–482. For a more detailed explanation of this point please refer to R3–C7.

We also acknowledge that the explanation of the accumulation periods selected for the SWEI was sub-optimal, potentially
70 causing confusion. As a result, it has been revised between lines 216–224 as follows:

*While this index has the potential to be applied across various time scales, its interpretation differs from that of other drought-related indices. This distinction is related to the presence of seasonal snowpack in specific regions, where several months during the warm season may exhibit a lack of SWE. The absence of SWE hinders the use of the SWEI during those months. As a result, Huning and AghaKouchak (2020) recommended to i) exclude the warm season months (in this case,*
75 *November to April) to prevent potential biases in the analysis of the SWEI, and ii) use accumulation periods shorter than 6 months for SWEI.*

*Therefore, we computed the SWEI for accumulation periods of 1, 3, and 6 months, assuming that the selected SWE dataset offers valuable insights into the spatial and temporal distribution of snow. Additionally, we filtered out grid cells with very low SWE ($<$1 mm) to avoid introducing bias in the calculation of the index.*

80 **R1–C5: Is it possible to use other linear or non-linear analyses to establish the relationship between the indices and SSI-1? Correlation analysis gives a first look at the relationships, but it is a linear version of the analyses. It may be that non-linear relationships that are possibly hidden in the treated data cannot be seen. Here a question arises: is the Event Coincidence Analysis (ECA) strong enough as an alternative method of assessing the relationship between droughts and their triggers? ¿Why?**
85 We appreciate this insightful comment. We used a linear cross-correlation to assess the relationship between SSI-1 and the other drought indices, following the approach of Barker et al. (2016); Peña-Gallardo et al. (2019a), although it is indeed possible to explore other statistical analyses (Fang et al., 2020; Guo et al., 2020) or other non-linear analyses, such as wavelets (Maity et al., 2016; Gyamfi et al., 2019; Li et al., 2020; Zhou et al., 2021). However, linear or non-linear analyses, including the cross-correlation, consider the relationship between the entire signal of the evaluated indices and the SSI-1, encompassing
90 periods of both surplus and deficits of their respective variables. As highlighted in our manuscript between lines 545–546, a high cross-correlation value does not necessarily imply that the evaluated indices could be used to assess streamflow drought conditions. To address this issue we added the Event Coincidence Analysis. This method, as described by Donges et al. (2016);

Siegmund et al. (2017) and detailed in our manuscript (lines 268–303), is specifically designed to evaluate the inter-dependency of events between two time series, focusing only on periods of deficits in the selected variables. Therefore, we consider it adequate to assess the relationship between meteorological, soil moisture, or snow drought indices and the SSI-1 during drought conditions. In the revised manuscript we have emphasised the importance of employing methods that specifically isolate drought conditions, as opposed to considering the entire time series of the indices, as indicated in lines 550–552: *Therefore, future studies evaluating the use of meteorological, soil moisture, and snow drought indices as proxies of streamflow drought should consider methods that evaluate solely drought conditions, such as the event coincidence analysis.*

**R1–C6: In the case of the ECA, ¿were the same thresholds selected for the several indices? That is, if moderate hydrological drought events were selected in the SSI-1 (SSI-1<-1.0), it was assumed that they were triggered by moderate meteorological/soil/snow events (e.g. SPI<-1.0, SPEI<-1.0, ...). ¿Do you consider merging the severities of hydrological and meteorological/soil/snow droughts? (e.g. that some severe droughts SSI-1<-1.5 are triggered by moderate meteorological droughts: -SPI<-1.0, or SPEI<-1.0-).**

We appreciate this comment. It is worth noting that we used the same predefined thresholds for all drought indices in the event coincidence analysis. This clarification has been added in lines 297–298 as follows: *Here, we use independently two different thresholds to define an event [...].*

While it is common in the literature to select predefined thresholds (e.g., Barker et al., 2016; Peña-Gallardo et al., 2019a; Bevacqua et al., 2021), we acknowledge that assessing the sensitivity of these thresholds for classifying drought events is an important area of study. However, conducting a comprehensive analysis of threshold sensitivity across multiple catchments and multiple indices is beyond the scope of this particular study.

To provide additional information regarding this aspect, we added the results on threshold values found by (Bachmair et al., 2015) giving a wider context to our results between lines L389-398:

*The SWEI scored the lowest results (median values lower than 0.4 and not statistically significant) among the evaluated indices, which is expected since the snow influence is confined to a limited number of catchments. Finally, the median coincidence rates for SWEI are all also lower than 0.4 and not statistically significant. This can be explained by the fact that the SWEI and ESSMI have fewer events below the $-1.5$ threshold used to define severe events compared to the SSI-1, which affects the characterisation of drought events. In other words, different indices might have different thresholds related to the occurrence of specific drought events. This is in agreement with Bachmair et al. (2015), who found that there is no single best threshold value for correctly identifying the onset of drought impacts, because impacts occur within a range of threshold values. This result suggests that the threshold used to define moderate or severe droughts for the case of the event coincidence analysis could vary according to the component of the hydrological cycle that is being evaluated; however, the evaluation of the optimal threshold to characterise droughts is out of the scope of this study.*

Finally, we have included this point in the prospects for future research in lines 577–578, stating: *Future studies could focus on ... iii) evaluating the sensitivity of the selected threshold to classify droughts.* This recognises the significance of a threshold sensitivity analysis as a potential topic for future investigations.

**R1–C7: Pag. 13, you say: While the analysis was done for diverse lags, the cross-correlation values and precursor coincidence rates, decreased gradually from zero lag (lag = zero months) to a lag of 12 months. ¿Is true that sentence for all the analyzed catchments? If the answer is "yes", ¿what do you think is the process that drive the flow in the catchments? The response is especially interesting in the nival and nivo-pluvial catchments, where the runoff should be delayed some weeks with respect to snow precipitation.**

Thank you for this remark. In general, the cross-correlation values and precursor coincidence rates, decreased gradually from zero lag (lag = zero months) to a lag of 12 months, but this did not happen for all analysed catchments. We acknowledge that it may seem like this is the case in our manuscript, and therefore, we have adjusted the following sentences:

1. *While the analysis was done for diverse lags, in general, the cross-correlation values and precursor coincidence rates, decreased gradually from zero lag (lag = zero months) to a lag of 12 months* (lines 313–315); and

2. *Figures S21–S24 and S25–S33 of the Supplementary Material show that the cross-correlation and event coincidence analysis values, in general, decreased gradually from zero lag to a lag of 12 months for SPI, SPEI and ESSMI at*

*all temporal scales, with the only exception of SPI and SPEI for nival catchments, where high correlation values and*
140  *coincidence rates extend to lags of 3-6 months* (lines L443–446).

The higher overall values of cross-correlation and event coincidence rates observed for almost all catchments at zero lag compared to lags exceeding 1 month could be attributed to the fact that the rapid response of the catchments actively contributes to assessing streamflow deficits. This influence is particularly pronounced in pluvial catchments compared to nival ones. This insight has been incorporated into the conclusions section, specifically between lines L561–563, which now reads as follows:
145  *In general terms, the cross-correlation and event coincidence analysis values decreased gradually from zero lag to a lag of 12 months. This could be explained by the fact that the rapid response of the catchments actively contributes to assessing streamflow deficits.*

**Reviewer 2**

**R2–C1: This study offers a holistic methodology for comprehending streamflow droughts across diverse hydrological contexts through the utilization of numerous hydrological and meteorological indices across Chile. In general, the manuscript is well-written. Nonetheless, substantial revision is necessary to rectify numerous methodological flaws and improve the manuscript's readability.**

Thank you for your valuable feedback and for reviewing our manuscript. We appreciate your positive comments about the study's overarching methodology and the overall quality of the paper. We have carefully considered your suggestions regarding the methodological aspects and readability of the manuscript.

**R2–C2: Concerning the application of a one-month timescale to represent hydrological drought, I have several reservations. One potential drawback is that the temporal scale of one month may be exceptionally susceptible to short-term fluctuations in streamflow, which may result in an overestimation or underestimation of drought conditions. Furthermore, the index may be impacted by short-term anomalies caused by particular weather phenomena, which may introduce extraneous data and potentially obscure longer-term patterns of hydrological drought. In addition, the responses of hydrological processes to precipitation events, such as groundwater recharge and base flow contribution to streams, may be delayed. These delayed effects might not be captured on a one-month scale, resulting in an inadequate representation of streamflow conditions. Critically, a one-month scale may not sufficiently capture the seasonal dynamics of streamflow in catchments with substantial snowmelt contributions, particularly if the SSI is computed for months in which snowmelt does not predominate as a hydrological process.**

Thank you for your insightful comment. First and foremost, we would like to clarify that our analysis explicitly focuses on streamflow drought, and thus, groundwater recharge is not within the scope of this study. However, we recognise the concern raised by the reviewer regarding the one-month temporal scale, which can be influenced by short-term fluctuations in streamflow, although less so than precipitation-based indices (see Figure 1 in R1-C2). We explain further our choice of the SSI-1 timescale in our response to R1-C2.

Furthermore, we agree with the reviewer's comment that a one-month scale may not fully capture the seasonal dynamics of streamflow in catchments with significant snowmelt, especially when the SSI is computed during months when snowmelt is not the dominant hydrological process. However, we believe this aspect is crucial for our analysis. The advantage of using the SSI-1 lies in its lower sensitivity to the annual seasonal cycle in the selected catchments. This aligns with our objective to assess how streamflow interacts with other components of the hydrological cycle. Aggregating streamflow into seasonal values could potentially obscure the relationships we are investigating, such as the influence of snowmelt on streamflow.

**R2–C3: Also, catchments characterized by intricate hydrogeology may exhibit non-linear correlations among precipitation, soil moisture, and streamflow. As such, a one-month SSI may not provide an accurate representation of real-world drought conditions.**

We partially agree with reviewer 2. In fact, not only catchments characterised by intricate hydrogeology but all catchments exhibit non-linear relationships between different fluxes of the hydrological cycle, and probably that is one of the reasons why there is no consensus how to monitor low flows operationally. However, even with all the recognised limitations of standardised drought indices, the SSI-1 has been widely used in literature to characterise streamflow droughts. Please refer to our reply to R1–C2 for further details.

**R2–C4: The sensitivity of the coincidence analysis between the SSI-1 and meteorological, soil moisture, and snow drought indices to the thresholds used to define each index remains uncertain.**

Thank you for this comment. As mentioned in our previous response to Reviewer 1 (R1–C6), we used predefined thresholds for all drought indices in the event coincidence analysis, and we acknowledge the importance of assessing threshold sensitivity. However, given the already wide scope of this study and the focus on multiple catchments and indices, a comprehensive analysis of threshold sensitivity is not within the scope of this particular work.

To underscore the significance of this aspect, we have included it in the prospects for future research in the manuscript, recognising the potential for future investigations to focus on evaluating the sensitivity of selected threshold values for classifying drought events in event coincidence analysis (lines 577-578).

**R2–C5: The selection of time scales and indices is significantly influenced by the accessibility and reliability of data, especially for soil moisture and snowpack, and streamflow data. Data may be scarce or unreliable in some catchments, thereby compromising the precision of drought assessments.**

We agree with Reviewer 2 on the importance of data availability and reliability for drought assessments and we consider that is is a common challenge in assessing droughts over data-scarce areas. In our study, we used state-of-the-art gridded datasets to characterise meteorological, soil moisture, and snow water equivalent-related droughts across the study area. This approach overcomes data scarcity issues for variables like soil moisture and snowpack in the evaluated catchments.

We acknowledge the inherent uncertainty in global and regional data products, and we paid careful attention in selecting the data products used to compute the selected drought indices. For example, we evaluated various soil moisture products, ultimately choosing ERA5-Land over alternatives like ERA5, SMAP-L3E, and SMAP-L4SM (please refer to Section 1 of our Supplementary Material). In the case of $SWE$, ERA5-Land was selected based on its superior cross-correlation values compared to ERA5. Please note that we did not use GLOBSNOWv3.1 because its coverage is limited to the global north or AMRS-E due to its relatively short period of record.

Moreover, recognising the potential delay in the availability of on-site streamflow measurements, we used open-access, spatially-distributed, near real-time products related to precipitation, potential evaporation, soil moisture, and $SWE$ as proxies for streamflow drought assessment in data-scarce regions. To ensure the reliability of the streamflow observations, we used quality-controlled data from the Center for Climate and Resilience Research (CR2), as noted on lines 240–241 of the revised manuscript: *Quality-controlled Q data were obtained for each catchment from the Center for Climate and Resilience Research (CR2; http://www.cr2.cl/datos-de-caudales/).*

**R2–C6: In the same context, there is substantial variation in the extent to which snowmelt contributes to streamflow across catchments and years. In order to evaluate the effects of streamflow drought on a snow water equivalent (SWE) index, comprehensive calibration and validation procedures are necessary.**

We appreciate your insightful comment about variations in the contribution of snowmelt to streamflow across different catchments and years. Indeed, the inter-annual variability of snowpack can substantially influence streamflow volumes in snow-influenced catchments. This consideration was a driving factor behind the inclusion of the Snow Water Equivalent Index (SWEI) in our analysis. However, it is important to note that our analysis does not intend to evaluate the effects of streamflow droughts on the SWEI. Instead, our study focuses on the reverse relationship, exploring how readily available meteorological, soil moisture, and snow drought indices impact streamflow droughts within the catchments.

Furthermore, it is worth noting that our approach does not involve a hydrological modelling framework, which means calibration and validation procedures are not applicable. We rely on statistical analyses to investigate the interactions between readily available drought indices and streamflow drought. We hope this clarification addresses your concern and provides a better understanding of our methodology.

**R2–C7: The intricate correlation between streamflow and soil moisture is susceptible to the impact of vegetation and soil properties. This intricacy may not be comprehensively encapsulated by a single drought index.**

We agree that the complex relation between streamflow and soil moisture is influenced by vegetation and soil properties. However, analysing the impact of these factors on this relationship is beyond the scope of this paper. Moreover, despite the acknowledged limitations of standardised drought indices, the SSI-1 remains a widely used measure in literature for characterising streamflow droughts. For more details, please see our response to R1–C2.

**R2–C8: I suppose the selected catchments are irregular in terms of total area and dominant hydrological and meteorological conditions. This may imply that hydrological regimes within a given catchment may vary considerably, which makes it difficult for single index or specific time scale to represent an entire catchment.**

We would like to address a potential misunderstanding. The hydrological regimes were specifically assigned to each catchment individually. Consequently, each of the 100 evaluated catchments corresponds to one unique hydrological regime. In the initial version of the manuscript, readers were directed to Baez-Villanueva et al. (2021) for a detailed explanation of this classification process. In the revised manuscript, between lines 124–131, we have explicitly outlined how the hydrological regimes were determined:

240  *As explained in (Baez-Villanueva et al., 2021), we classified these catchments into four hydrological regimes: i) nival (snow-dominated), ii) nivo-pluvial (snow-dominated with rain influence), iii) pluvio-nival (rain-dominated with snow influence), and iv) pluvial (rain-dominated regime). The CAMELS-CL dataset provides extensive information on 516 Chilean catchments, including hydroclimatic indices, geomorphological characteristics, location, topography, geology, soil types, land cover, hydrological signatures, and the degree of human intervention. These hydrological regimes were derived by analysing P and Q*
245  *data, and the timing of Q peaks relative to maximum P. Finally, the derived hydrological regimes were compared against other Studies (Baez-Villanueva et al., 2021).*

Finally, regarding the differences in terms of total area and dominant hydrological and meteorological conditions, please refer to our response to R2-C9.

**R2–C9: The justification for the selection of these particular 100 catchments should be clarified better. It is necessary**
250  **to include a table or figure that compares the various hydroclimatological and land use characteristics of the chosen catchments.**

Thank you for this comment. In the revised manuscript, we have expanded on the rationale for selecting the catchments used in our study, including the selection criteria. These changes can be found between lines 106–124 and are as follows:

*We carefully selected a subset of 100 near-natural catchments, as detailed in Baez-Villanueva et al. (2021), aiming to*
255  *isolate the influence of natural hydrological processes from human interventions. By selecting catchments with minimal human alterations, our goal was to more clearly understand the intrinsic relationships between meteorological, soil moisture, and snow drought indices and streamflow dynamics in a predominantly natural context. The selected catchments met the following criteria:*

260  1. *Less than 25% of missing values in the daily Q time series (which could be non-consecutive).*

2. *Absence of large dams.*

3. *Less than 10% of Q allocated to consumptive uses (intervention degree < 10%).*

4. *Catchments not dominated by glaciers (glacier area < 5%).*

5. *Urban area less than 5%.*

265  6. *Minimal irrigation abstractions (agriculture fraction < 20%).*

7. *Less than 20% of the area covered by forest plantations.*

8. *No signs of artificial regulation in the hydrograph (resulting in the exclusion of 10 catchments).*

*These selected catchments vary in size (35 to 11,137 km$^2$, median area of 645 km$^2$), dominant Köppen-Geiger climate type (from BWh, BWk, BSh, BSk or ET), annual precipitation (from 56.58 to 3,914.26 mm y$^{-1}$), aridity index (from 0.28 to 9.32),*
270  *dominant land cover type (shrublands, grasslands, barren land, or native forests) and main geological characteristics. They are part of the Catchment Attributes and Meteorology for Large-sample Studies dataset in Chile (CAMELS-CL; Alvarez-Garreton et al. (2018)), and their attributes are summarised in the Supplementary Material.*

Additionally, we have expanded our Table S6 in the Supplement Material to include key features such as the catchment area, the dominant land cover and its percentage, the aridity index, mean annual streamflows, and a figure depicting mean monthly
275  precipitation and streamflows.

**R2–C10: The results are not adequately discussed in relation to the existing literature concerning Chile and beyond. It is my belief that introducing the "Discussion" and "Results" sections independently is crucial.**

We thank Reviewer 2 for these comments. Regarding the discussion of our results in relation to the existing Chilean and international literature, please see our responses to R1–C2, R3–C4, and R3–C5 for further details. However, we prefer not
280  to separate the Discussion and Results sections. In the first draft of the manuscript, we initially separated these sections, but

found that to maintain a self-contained Discussion, we had to repeat numerous short text segments. Consequently, we decided to combine both sections into one, aiming to reduce the overall length of the manuscript and to improve the clarity of the discussion about the results.

**R2–C11: The title is considerably lengthy. It should be revisited to be more concise and informative.**

We acknowledge that the title was quite lengthy. We have since shortened it, and the revised title now reads as follows: *On the time scales of drought indices to monitor streamflow drought over catchments with different hydrological regimes*.

**R2–C12: In the abstract, it is suggested to emphasize the importance of the research outcomes and the possible ramifications they may have on the domains of hydrology or drought monitoring. It can also be enhanced by presenting quantitative results.**

Thank you for your comment. We have enhanced the clarity and significance of our research in the revised abstract. We drought explicitly highlight the differences between nival and pluvial catchments, presenting specific details on the index that performed the best, accompanied by numerical values (please refer to R3–C2). Additionally, we agree with Reviewer 2 regarding the importance of explicitly addressing the potential implications of our study on the fields of hydrology and drought monitoring. This consideration was already incorporated into the outlook paragraph at the end of the conclusion. Given the current length of the abstract, we prefer not to include this information within it.

**R2–C13: Prevent "lump" referencing; that is, do not refer excessively to illustrate a point or provide a description in the paper (e.g., L22-24).**

We have carefully revised the manuscript and addressed your concerns. Specifically, we have removed references that were deemed excessive for illustrating a point, as suggested, particularly in lines 24 and 59. We believe these changes have strengthened the clarity and focus of the manuscript.

**R2–C14: The in-text citation format must be meticulously revised in accordance with the journal's specifications.**

We used the LaTeXtemplate provided by the journal for manuscript preparation, and we may not fully grasp the reviewer's concern behind this comment. Nevertheless, we have meticulously revised the in-text references as suggested. In line with this, we have corrected a typo related to the placement of the parentheses in line 502 and organised the references mentioned in lines 166, 184, and 234 chronologically. We believe these revisions have enhanced the overall quality of the manuscript. If further clarification is needed regarding the use of the journal's template, we would appreciate if the reviewer could provide explicit examples.

**R2–C15: L37: Please include agricultural and socioeconomic droughts.**

Agricultural, socio-economic, and ecological droughts are now introduced in the revised manuscript between lines L38–39.

**R2–C16: L60: "climate factors" <> "climate conditions".**

The term "climate factors" has been updated to "climate conditions" in lines 61–62, and the revised sentence now reads as follows: *This propagation may lead to clustered, attenuated, delayed, and prolonged events, which are influenced by catchment characteristics, climate conditions, and soil moisture dynamics[...].*

**R2–C17: In the text, SWE and SWEI are utilized interchangeably.**

Thank you for your comment. After a thorough review of the manuscript, we believe that both terms are used appropriately throughout the manuscript. $SWE$ refers to snow water equivalent, and is introduced in line 55; while the standardised index based on $SWE$ is referred to as the standardised Snow Water Equivalent Index (SWEI), as documented in Huning and AghaKouchak (2020), and it is introduced in line 9.

To avoid any potential ambiguity in the manuscript concerning the use of both terms, we have replaced the phrase "... snow (SWEI) drought indices..." with "... snow-related (SWEI) drought indices..." in the sentence at lines 141–142, which now reads as follows: *"...computing meteorological (SPI, SPEI), soil moisture (ESSMI), and snow-related (SWEI) drought indices[...]".*

**R2–C18: The explanation is absent for Equation 3.**

We would like to express our gratitude to Reviewer 2 for identifying this omission in our manuscript. We have addressed this issue by incorporating the explanation of Equation 3 in the revised manuscript, which now reads as follows: *where $\bar{X}$ represents* the mean value of the SSI-1, and $\bar{Y}$ represents the mean value of the SPI, SPEI, ESSMI, and SWEI time series, alternatively.

325

**R2–C19: Equation's numbering needs to be revised.**

We have conducted a comprehensive review of the equation numbering, and they appear correct to us. We are using the LATEXtemplate provided by HESS, which automatically enumerates equations and facilitates their cross-referencing.

**Reviewer 3**

**R3–C1: Overall, the presented research is a solid and valuable addition to drought science, posing relevant questions for drought management and monitoring. The introduction is well-structured, offering a comprehensive review of drought indices. The methods and results are clearly articulated. I have only a few general and minor comments to offer for further enhancement.**

We sincerely thank Reviewer 3 for recognising the value of our work as a solid contribution to drought science. We appreciate the thoughtful comments provided, which we believe have significantly enhanced the overall clarity and understanding of the manuscript.

**R3–C2: While the science questions are well-framed and articulated, the takeaways in the abstract seem somewhat general. The authors could enhance these by distilling the results into more compelling and informative points. Focusing on 'what/why' questions rather than providing simple 'yes/no' answers.**

We express our gratitude to Reviewer 3 (and Reviewer 2) for their insightful suggestions regarding the abstract . As a response, we have enhanced the visibility of the results in our abstract. Specifically, we now explicitly address the distinctions between nival and pluvial catchments, providing details on which index performed the best, complete with numerical values. The revised abstract is as follows:

*There is a wide variety of drought indices, yet a consensus on suitable indices and temporal scales for monitoring streamflow drought remains elusive across diverse hydrological settings. Considering the growing interest in spatially-distributed indices for ungauged areas, this study addresses the following questions: i) what temporal scales of precipitation-based indices are most suitable to assess streamflow drought in catchments with different hydrological regimes?, ii) do soil moisture indices outperform meteorological indices as proxies for streamflow drought?, iii) are snow indices more effective than meteorological indices for assessing streamflow drought in snow-influenced catchments? To answer these questions, we examined one hundred near-natural catchments in Chile with four hydrological regimes, using the Standardised Precipitation Index (SPI), Standardised Precipitation and Evapotranspiration Index (SPEI), Empirical Standardised Soil Moisture Index (ESSMI), and standardised Snow Water Equivalent Index (SWEI), aggregated across various temporal scales. Cross-correlation and event coincidence analysis were applied between these indices and the Standardised Streamflow Index at a temporal scale of one month (SSI-1), as representative of streamflow drought events. Our results underscore that there is no a single drought index and temporal scale best suited to characterise all streamflow droughts in Chile, and their suitability largely depend on catchment memory Specifically, in snowmelt-driven catchments characterised by a slow streamflow response to precipitation, the SPI at accumulation periods of 12–24 months serves as the best proxy for characterising streamflow droughts, with median correlation and coincidence rates of approximately 0.70–0.75 and 0.58–0.75, respectively. In contrast, the SPI at a 3-month accumulation period is the best proxy over faster-response rainfall-driven catchments, with median coincidence rates of around 0.55. Despite soil moisture and snowpack being key variables that modulate the propagation of meteorological deficits into hydrological ones, meteorological indices are better proxies for streamflow drought. Finally, to exclude the influence of non-drought periods, we recommend using the event coincidence analysis, a method that helps assessing the suitability of meteorological, soil moisture, and/or snow drought indices as proxies of streamflow drought events.*

**R3–C3: The method section notes that the soil moisture and snow drought index are calculated using a non-parametric approach, in contrast to the parametric method used for other drought indices. I suggest the authors discuss the implications of using different methods. It would be insightful to conduct analysis demonstrating the uncertainties in drought assessment that arise from using various approaches and distribution functions.**

We appreciate this insightful comment. The indices employed in our study have been drawn from well-established literature, have been widely used, and their developers have extensively explored diverse distributions and approaches themselves, which contributes to the robustness and reliability of the selected indices.

While we acknowledge the importance of a detailed exploration into the selection of a specific approach or distribution, we would like to emphasise that such an analysis is beyond the scope of the current study and could be a subject for a dedicated article. Nevertheless, we recognise the value in providing readers with insights into how the choice between parametric and non-parametric approaches may influence the analysis.

375      To address this, we computed a standardised soil moisture index using both a parametric approach with the gamma distribution and the non-parametric approach proposed by Carrão et al. (2016) (ESSMI) across all catchments. As expected, there are variations in the values computed through applying both approaches. Figure 2 illustrates these differences, computed as the *parametric approach* minus the *non-parametric approach*, focusing on a nival and a pluvial catchment.

[Figure]

**Figure 2.** Differences between a standardised soil moisture index calculated with a parametric (gamma distribution) and non-parametric (ESSMI) approach for scales ranging from 1 to 12 months. Panel *a* shows these differences over a Nival catchment, while panel *b* over a pluvial catchment.

375      This figure has now been included in the Supplementary Material, and we included the fact that drought indicators computed
380 with parametric and non-parametric approaches could yield different results between lines 569–572 and reads: *Additionally, we used both parametric and non-parametric approaches to compute the drought indices, aligning with the methods established in their respective literature. However, the selection between parametric and non-parametric approaches can impact the computed values of the respective indices (Soláková et al., 2014; Mallenahalli, 2020) and their associated drought characteristics (Tijdeman et al., 2020a). To illustrate this, Figure S34 of the Supplement Material shows the differences of a standardised soil*
385 *moisture index using both a parametric approach with the gamma distribution and the non-parametric approach proposed by (ESSMI Carrão et al., 2016) focusing on a nival and pluvial catchment.*

      Additionally, we included between lines 575–576 that further research can focus on evaluating the uncertainties related to the selection of parametric or non-parametric approaches for calculating diverse drought indices across multiple temporal scales.

**R3–C4: In the Results section, before the correlation analysis, adding a section that gives an overview of the drought characteristics in the region would provide useful context. This overview should include aspects such as the seasonality of drought, as well as the spatial distribution, frequency, and severity of droughts based on different indices.**

We thank Reviewer 3 for raising this point. We agree that adding a section that gives an overview of the drought characteristics in the region would provide useful context. However, given the spatial heterogeneity of drought events in the study area and the amount of indices and temporal scales that we selected, discussing the spatial distribution, frequency and severity of drought events in Chile is beyond the scope of this article, and it could be the subject of a different article.

However, in order to help the understanding of the general characteristics of drought events in Chile, we refer the reader to the most relevant references in the following new text:

Between lines 99–105: *Boisier et al. (2016) found that large-scale circulation changes and the Pacific Decadal Oscillation explains around 50% of the P decline observed in Central Chile, while the Antarctic stratospheric ozone depletion has played a major role in the summer P decline Boisier et al. (2018). On the other hand, Garreaud et al. (2017) analysed the extraordinary character of the megadrought using century long historical records and a millennial tree-ring reconstruction of regional P, along with describing its impacts on regional hydroclimate and vegetation, while Garreaud et al. (2019) found that the exceptional length of the megadrought is due to the prevalence of a circulation dipole-hindering the passage of extratropical storms over Central Chile.*

And between lines 134–139: *Recently, Alvarez-Garreton et al. (2021) found that annual P anomalies during the megadrought have been larger in nival catchments compared to pluvial catchments, with nival catchments being more prone to an intensified propagation of persistent droughts. This was attributed to the accumulation of precipitation deficits on basins with longer streamflow response time to precipitation (refer as catchment memory), compared to fast response pluvial basins.*

**R3–C5: This study concentrates on the CAMELs headwater catchments. I recommend the authors provide discussions from regional perspectives, e.g., upscaling the catchment-scale insights to a wider regional context.**

We thank Reviewer 3 for this comment. It is true that comparison of our results with existing literature is limited to Alvarez-Garreton et al. (2018) and Barker et al. (2016), which was due to the lack of other articles analysing drought indices for catchments with different hydrological regimes. We now added the following discussion:

Between lines 344–358: *It is worth noting that the highest values in Central Chile (32–36°S) are obtained with SPI-12 ($\sim$0.6–0.8) and SPEI-12 ($\sim$0.4–0.8), indicating that streamflow droughts are related to longer P deficits over those catchments, which could be attributed to higher baseflow coefficients (Alvarez-Garreton et al., 2018, their Figure10e) and the existence of important aquifers in the central valley (Taucare et al., 2024). This is consistent with previous studies where the SPI-12 shows high correlation with groundwater droughts (e.g. Bloomfield and Marchant, 2013; Bloomfield et al., 2015; Folland et al., 2015; Apurv et al., 2017). On the other hand, the different temporal scales suggested in Table 2 for nival and nivo-pluvial catchments agree with result obtained by Peña-Gallardo et al. (2019b) for 289 catchment in the conterminous U.S. They found an important spatial heterogeneity in hydrological droughts of mountainous river basins, whose response was controlled not only by precipitation variability but also by temperature. Finally, and even though the methodologies are not directly comparable, the temporal scales suggested for SPI and SPEI in Table 2 for all hydrological regimes but nival, partially agree with those of Meresa et al. (2023). They found that the overall lag time between the start of the meteorological and hydrological drought is almost 4–6 months, using SPI and SPEI at different temporal scales for analysing nine subcatchments of the Awash basin in Ethiopia, without distinction of the hydrological regime.*

Moreover, to emphasise the wide diversity of catchment characteristics of the 100 catchments selected for this study, which should help the wider audience to get catchment-scale insights beyond the CAMLES-CL dataset, the following new paragraphs were added:

Between lines 120–122: *These selected catchments vary in size (35 to 11,137 km$^2$, median area of 645 km$^2$), dominant Köppen-Geiger climate type (from BWh, BWk, BSh, BSk or ET), annual precipitation (from 56.58 to 3,914.26 mm y$^{-1}$), aridity index (from 0.28 to 9.32), dominant land cover type (shrublands, grasslands, barren land, or native forests) and main geological characteristics. They are part of the Catchment Attributes and Meteorology for Large-sample Studies dataset in Chile (CAMELS-CL; Alvarez-Garreton et al. (2018)), and their attributes are summarised in the Supplementary Material.*

Between lines 499-501: *The drought indices are presented in Table 2. These drought indices were chosen to accommodate the diversity of the catchments.*

**R3–C6: Please define some technical terms that may not be commonly understood by all readers. For example, clarifying the meaning of 'nival hydrological regime'.**

We appreciate your suggestion to provide clear definitions for technical terms that might not be clear to all readers. Consequently, we have defined the terms 'nival and 'pluvial' in the revised manuscript between lines 124–126: *As explained in (Baez-Villanueva et al., 2021), we classified these catchments into four hydrological regimes: i) nival (snow-dominated), ii) nivo-pluvial (snow-dominated with rain influence), iii) pluvio-nival (rain-dominated with snow influence), and iv) pluvial (rain-dominated regime).*

And between lines 131–133: *The nival hydrological regime characterises catchments dominated by snowfall, where the peak Q occurs in spring or summer due to snowmelt. In contrast, the pluvial regime characterises catchments dominated by rainfall, where Q is driven by rainfall-runoff processes.*

**R3–C7: given that the correlation between SWEI and SSI are generally low (75th percentile < 0.4), the following statement lacks solid grounding: "snow accumulated during the cold season influences the Q observed in the next hot season." And later in Line 395: "The SWEI showed the highest values over nival catchments, followed by nivo-pluvial and pluvio-nival catchments, respectively". While true in a comparative sense, the performance of SWEI in nival catchments is still lower than that of SPI. In general, with the results presented, it is not a strong argument that snow drought is critical for Q drought even for catchments with seasonal snowpack. Given the presented results, it is not a convincing argument to suggest that snow drought is a critical factor for discharge drought, even in catchments with seasonal snowpack. A more detailed examination of historical snow droughts in this region could offer a better perspective. Investigating whether severe snow droughts are a rare occurrence in this region and, consequently, whether they are not a reliable indicator for discharge droughts, may be able to provide a more robust conclusion.**

We thank the reviewer for this comment. Indeed, the initial version of the manuscript unintentionally gave the impression that the SWEI performed better than it actually did. In the revised manuscript, we have made the following changes to clarify this:

1. *snow accumulated during the cold season has a relatively small influence on Q observed in the subsequent warm season.* (lines 330–331);

2. *While the SPI, SPEI, and ESSMI generally exhibited higher values across all regimes, the SWEI showed relatively high values in nival catchments (75th percentile < 0.51). This was followed by nivo-pluvial (75th percentile < 0.49) and pluvio-nival (75th percentile < 0.28) catchments.* (lines 455–457); and

3. Additionally, we replaced *"the highest values"* with *"relatively high values"* between lines 483–484. Now, the sentence reads: *The SWEI showed relatively high values in pluvio-nival catchments [...].*

To further emphasise that the SWEI may not be a decisive factor in assessing $Q$ droughts, we have added the following statement between lines 489–492: *Additionally, these results suggest that the SWEI alone is not capable of representing Q droughts in snow-influenced catchments, which could be due to ERA5-Land's limitations in representing SWE in these areas. However, combining it with other indices might be beneficial, which is in line with previous studies highlighting the importance of incorporating snow-related information in drought analysis and prediction (Huning and AghaKouchak, 2020; Gottlieb and Mankin, 2021).*

---

## Author Response (AR1)

**Comments to Editor – HESS-2023-1911**

Oscar M. Baez-Villanueva, Mauricio Zambrano-Bigiarini, Diego G. Miralles, Hylke E. Beck, Jonatan F. Siegmund, Camila Alvarez-Garreton, Koen Verbist, René Garreaud, Juan Pablo Boisier, Mauricio Galleguillos

Hydro-Climate Extremes Lab (H-CEL)

Ghent University, Ghent, Belgium
* * *
E-mail: obaezvil@gmail.com

December 31st, 2023

**Dr. Anne Van Loon**

**Handling Editor**

**Hydrology and Earth System Sciences**

Dear Editor,

We extend our sincere appreciation for your efficient handling of our manuscript and for providing insightful and concise feedback following our response to the reviewers. In the subsequent paragraphs, we provide a summary of the changes implemented in the manuscript based on your valuable suggestions. Changes addressed in the response to reviewers document, not explicitly mentioned here, have also been incorporated into the revised manuscript. Thank you for your support and for helping us improving the manuscript :)

Additionally, we would also like to sincerely thank you for handling the manuscript in such a prompt and efficient manner.

Sincerely,

Oscar Manuel Baez-Villanueva

(on behalf of all authors)

**Editor comments**

**E–C1: SSI (reviewer 1 & 2): Two of the reviewers commented on the use of the time scale for SSI. I agree with your rebuttal that there is no need to accumulate SSI because it reflects streamflow and therefore integrates catchment processes (including groundwater discharge and snow accumulation and melt). I do suggest to make the suggested changes in the manuscript to clarify this also for the reader (including the additional references).**

All changes described in the response to the reviewers (R1–C2, R2–C2, R2–C3) have now been added to the revised manuscript.

**E–C2: Zero lag (reviewer 1): I do not agree with your explanation that the lack of lag is related to a rapid catchment response. Catchment storage delays are already included in the accumulation period, which is a better indicator for rapid or slow response. Only processes that cause a strong temporal shift in response (like snow accumulation and melt) will show a clear lag. An optimal lag of zero was also found in other studies, which shows that shifts generally do not occur, even in slowly-responding catchments. Please reformulate your suggested revision of the text.**

Thanks for the clarification! The text in the manuscript has been reformulated in L567–570 to:

*In general terms, the cross-correlation and event coincidence analysis values decreased gradually from a lag of zero months to a lag of 12 months. This could be explained by the fact that only processes that cause a strong temporal shift in response (like snow accumulation and snowmelt) will show a clear lag response. An optimal lag of zero months was found by (Peña-Gallardo et al., 2019), which indicates that these shifts generally do not occur, even in slowly-responding catchments.*

**E–C3: SWE (reviewer 1 & 3): thanks for the explanation. I agree with this, but I have a fundamental issue here. The SWEI indicates a lack of snow ACCUMULATION, but what is important for streamflow is snow MELT. The effects of low snow accumulation are only noticeable in streamflow during the melt season, so low correlations between SWEI and SSI are expected. I would therefore suggest to use the Standardized Snow Melt and Rain Index (SMRI) by Staudinger et al. (2014) instead of the SWEI. Using SMRI might also change the conclusion that snow is not so important (reviewer 3).**

Thank you for your comment. Indeed, the SWEI does not explicitly consider snowmelt, and we acknowledge that the SMRI could provide insights into how precipitation and snowmelt contribute to streamflow droughts. However, we believe that incorporating it might potentially alter the article's scope, including its objectives and overall narrative. Currently, we recommend indices and timescales that could be applied to ungauged catchments since precipitation ($P$), potential evaporation ($PE$), snow water equivalent ($SWE$), and soil moisture data can be obtained from openly available gridded datasets. On the contrary, the SMRI proposed by Staudinger et al. (2014) relies on a snowmelt model that requires a calibration strategy in each basin, making its application challenging over data-scarce settings.

To be transparent with our readers, we have included the following paragraph in L576–583:

*The SWEI standardises the time series of SWE, indicating a lack of snow accumulation. The low correlation values and event coincidence rates observed in the evaluation of the SWEI may be also attributed to the fact that the SWEI does not explicitly account for snowmelt and its impact on streamflow conditions. The effects of low snow accumulation become noticeable in streamflow only during the melt season; hence, lower correlations between the SWEI and SSI are expected. For future studies aiming to assess the influence of snowmelt on streamflow drought, we recommend using the Standardised Snow Melt and Rain Index (SMRI; Staudinger et al., 2014) instead of the SWEI. The SMRI requires the implementation and calibration of a snow model capable of representing snow processes in the target catchments.*

**E–C4: Thresholds (reviewer 1 & 2): I agree with you that a sensitivity analysis of thresholds is beyond the scope of the study. But, as you mentioned in the rebuttal, it is helpful to discuss the uncertainty related to thresholds and include it as suggestion for future research.**

All changes related to this point including the uncertainty related to the thresholds and the inclusion of the evaluation of the sensitivity of different thresholds for classifying drought events have now been added to the revised manuscript and are described in the response to the reviewers (R1–C6, R2–C4).

**E–C5: Parametric & non-parametric drought indices (reviewer 3): thanks for doing the initial analysis. I understand that doing a full analysis of the effect of different parametric and non-parametric analysis is beyond the scope of this study, but I do agree with the reviewer that it potentially increases uncertainty if you calculate correlations between indices that are calculated with different methods. I was for example triggered by your comments that not all indices have values lower than -2 (p.11 l.255) and "that the SWEI and ESSMI have fewer events below the -1.5 threshold" (p.16 l.335). This shows that the distribution of the values of the indices is not the same between the indicators and this affects your conclusions of the coincidence of drought events. You should discuss what the effects of this methodological choice on the overall results are.**

All changes related to this point that are described in the response to the reviewers (R3–C3) have now been added to the revised manuscript. Additionally, to clearly mention the impacts of this choice we have added in L589–594 the following statement:

*The combination of parametric and non-parametric approaches to classify drought events can potentially increase the uncertainty in the evaluation, as the distribution of standardised values may differ among indicators. The selection of parametric or non-parametric approaches could impact the occurrence of extreme events and, consequently, the correlation values and event coincidence rates. For instance, in the case of the non-parametric ESSMI and SWEI, less extreme values were observed compared to the SPI and SPEI parametric approaches.*

**E–C6:Furthermore, I agree with you that the Results and Discussion do not need to be separated (reviewer 2), but I do suggest to add more general discussion about what insights this research brings on a more general level (including references), either in the Results or in the Conclusion section. I also would suggest to shorten the title even further. And finally I don't see the need to mention agricultural and socioeconomic droughts (reviewer 2).**

A more general discussion about what insights this research brings on a more general level has been now included in the Conclusion Section between lines L552–566:

*Mountainous catchments play a pivotal role in supplying freshwater resources to both society and ecosystems. They contribute substantial runoff, redistribute winter P to spring and summer, and mitigate flow variability in lowlands (Viviroli and Weingartner, 2004; Viviroli et al., 2011). Numerous studies have demonstrated the impact of temperature increases and changes in snowfall fraction on droughts in mountainous areas (Marke et al., 2018; Blahušiaková et al., 2020), potentially affecting both the total volume of streamflow and its seasonality. Given that mountainous catchments generally remain ungauged, it is crucial to identify which drought indicators and timescales of monitored variables can be used to assess streamflow drought conditions. The approach demonstrated in this paper highlights the effectiveness of using individual drought indices to characterise streamflow drought, while also highlighting the varying behaviour of these indices across catchments with diverse hydrological regimes. This suggests that combining different drought indices based on catchment-specific responses (e.g., catchment memory Alvarez-Garreton et al., 2021) could provide further avenues for strengthening the relationship between meteorological, soil moisture, and snow drought indices as proxies of streamflow drought. Although catchment characteristics are important in understanding the relationship between meteorological, soil moisture, snow drought and streamflow droughts, this article provides a complementary perspective on how this relationship can also be related to the hydrological regime of the catchments. Our findings indicate that hydrological regimes offer valuable insights into the time lag at which these indices could be used to assess streamflow drought.*

Also, the title has been shorten and now reads: *On the timescale of drought indices for monitoring streamflow drought considering catchment hydrological regimes*. Finally, we have removed the references to both agricultural and socioeconomic droughts :)

**References**

Alvarez-Garreton, C., Boisier, J. P., Garreaud, R., Seibert, J., and Vis, M.: Progressive water deficits during multiyear droughts in basins with long hydrological memory in Chile, Hydrology and Earth System Sciences, 25, 429–446, https://doi.org/10.5194/hess-25-429-2021, 2021.

Blahušiaková, A., Matoušková, M., Jenicek, M., Ledvinka, O., Kliment, Z., Podolinská, J., and Snopková, Z.: Snow and climate trends and their impact on seasonal runoff and hydrological drought types in selected mountain catchments in Central Europe, Hydrological Sciences Journal, 65, 2083–2096, 2020.

Marke, T., Hanzer, F., Olefs, M., and Strasser, U.: Simulation of past changes in the Austrian snow cover 1948–2009, Journal of Hydrometeorology, 19, 1529–1545, 2018.

Peña-Gallardo, M., Vicente-Serrano, S. M., Hannaford, J., Lorenzo-Lacruz, J., Svoboda, M., Domínguez-Castro, F., Maneta, M., Tomas-Burguera, M., and El Kenawy, A.: Complex influences of meteorological drought time-scales on hydrological droughts in natural basins of the contiguous Unites States, Journal of Hydrology, 568, 611–625, 2019.

Staudinger, M., Stahl, K., and Seibert, J.: A drought index accounting for snow, Water Resources Research, 50, 7861–7872, 2014.

Viviroli, D. and Weingartner, R.: The hydrological significance of mountains: from regional to global scale, Hydrology and earth system sciences, 8, 1017–1030, 2004.

Viviroli, D., Archer, D. R., Buytaert, W., Fowler, H. J., Greenwood, G. B., Hamlet, A. F., Huang, Y., Koboltschnig, G., Litaor, M., López-Moreno, J. I., et al.: Climate change and mountain water resources: overview and recommendations for research, management and policy, Hydrology and Earth System Sciences, 15, 471–504, 2011.

---

## Author Response (AR2)

**Comments to Editor – HESS-2023-1911**

Oscar M. Baez-Villanueva, Mauricio Zambrano-Bigiarini, Diego G. Miralles, Hylke E. Beck, Jonatan F. Siegmund, Camila Alvarez-Garreton, Koen Verbist, René Garreaud, Juan Pablo Boisier, Mauricio Galleguillos

Hydro-Climate Extremes Lab (H-CEL)

5    Ghent University, Ghent, Belgium
* * *
E-mail: obaezvil@gmail.com

December 31st, 2023

**Dr. Anne Van Loon**

10   **Handling Editor**

**Hydrology and Earth System Sciences**

Dear Editor,

We extend our sincere appreciation for your efficient handling of our manuscript and, once again, for providing insightful and concise feedback following our response to your comments. In the subsequent paragraphs, we provide a summary of the
15   changes implemented in the manuscript based on your valuable suggestions. Thank you for your support and for helping us improve the manuscript :)

Sincerely,

Oscar Manuel Baez-Villanueva

(on behalf of all authors)

**Editor comments**

**E–C1: Thanks for the revisions. I think the manuscript has improved in terms of clarity and discussion.**

We appreciate your insightful feedback, and we are pleased to hear that the revisions have enhanced the clarity and discussion in the manuscript.

**E–C2: I understand your point about the SMRI. However, I do think that snow needs to be considered more carefully in your study. You mention in the revised manuscript that: "Finally, the median correlation values for SWEI are all lower than 0.4, with the highest values obtained for SWEI-1 at lags from zero to five months, suggesting that snow accumulated during the cold season has a relatively small influence on Q observed in the subsequent warm season" (l.344-345; in the tracked changes version of the manuscript) And in the conclusion that "meteorological drought indices are better proxies of streamflow drought events than SWEI index over snow-influenced catchments". (l.562-563). I think these statements are a bit misleading. In Figure S24 in the Supplementary Material there is an increase in correlation with SWEI lag times around 3–6 months for the nival catchments with median values above 0.5 (which is higher than the SPI1 and SPEI1 correlations for the same group of catchments). You mention this in Section 4.2: "SWEI showed relatively high values in nival catchments" (l.472), but it is a bit hidden. I would therefore strongly suggest that you rephrase the conclusion to state that for snow-dominated catchments, snow-accumulation and -melt processes are important drivers of streamflow, as can be seen by the high correlation of SSI with the snow accumulation index lagged 3-6 months or the precipitation-based indices accumulated over 6 months.**

We agree with the Editor that the statements mentioned earlier may create confusion. Regarding the first sentence (L328–331), please refer to EC–C3, where we have clarified that the median correlation values of the SWEI consistently average below 0.4 for all catchments.

Additionally, we have elaborated on the sentence found on L472 (L455–459 in the recently revised manuscript), which now reads:

*While the SPI, SPEI, and ESSMI generally exhibited higher values across all regimes, the SWEI showed relatively high values in nival catchments (see Fig. S24 of the Supplementary Material) for lags from 3 to 6 months (75th percentile $< 0.60$ and median values $\geq 0.50$). These values are higher compared to the P-based indices at the same temporal scales, indicating that snow accumulation and snowmelt processes are important drivers of Q.*

Finally, guided by the insights from Figure S24, we have rephrased the conclusion as recommended, and it now reads:

*Snow-dominated catchments have a larger memory and consequently, larger temporal scales of the drought indices can be used as proxy of streamflow drought. In general terms, snow accumulation and snowmelt processes are important drivers of Q. This influence is evident in the relatively high cross-correlations observed between the SSI and the snow accumulation index lagged 3–6 months or the P-based indices accumulated over 6 months.*

**E–C3: I also suggest some minor changes in these statements:**
**- "Finally, the median correlation values for SWEI are all lower than 0.4, with the highest values obtained for SWEI-1 at lags from zero to five months, suggesting that snow accumulated during the cold season has a relatively small influence on Q observed in the subsequent warm season" (l.344-345) » Here you need to add "on average for all catchments".**

**- "The larger time lags of SPI and SPEI over snow-dominated basins, as opposed to shorter lags in pluvial basins, are consistent with how catchment memory modulates the propagation of precipitation into the hydrological cycle, which in turn determine the time until a streamflow drought is influenced by the meteorological drought precursor." (l.480-483) » Here you need to mention snow, when you are talking about catchment memory, because in the nival catchments the catchment memory you are talking about is seasonal.**

Thank you for your suggestion. We have revised both sentences, and they now read as follows:

1. *Finally, the median correlation values for SWEI are all lower than 0.4 on average for all catchments, with the highest values obtained for SWEI-1 at lags from zero to five months. This suggests that, when considering all catchments collectively, snow accumulated during the cold season has a relatively small influence on the observed Q in the subsequent warm season. [L328–331]*; and

65     *2. The larger time lags observed for SPI and SPEI over snow-dominated basins, in contrast to the shorter lags in pluvial basins, along with the increased lag for SWEI, are consistent with how catchment memory modulates the propagation of solid and liquid precipitation into the hydrological cycle. This, in turn, determines the time until a streamflow drought is influenced by rainfall and snow-related processes (Alvarez-Garreton et al., 2021). [L465–468]*

    **E–C6:And finally some minor textual suggestions for you to change: - l.14-15: there is no a single drought index >**
70 **there is not a single drought index OR there is no single drought index - l.620: Supplement Material > Supplementary Material**

    Thank you for your textual suggestions :) The sentences have been revised as follows:

    1. *... there is not a single drought index ...*, and

    2. *... Supplementary Material...*

**75 References**

Alvarez-Garreton, C., Boisier, J. P., Garreaud, R., Seibert, J., and Vis, M.: Progressive water deficits during multiyear droughts in basins with long hydrological memory in Chile, Hydrology and Earth System Sciences, 25, 429–446, https://doi.org/10.5194/hess-25-429-2021, 2021.